# A unified model-based framework for doublet or multiplet detection in single-cell multiomics data

Haoran Hu [1], Xinjun Wang [2], Site Feng [3,4], Zhongli Xu [4,5], Jing Liu[5], Elisa Heidrich-O'Hare[3], Yanshuo Chen[6,7], Molin Yue[1], Lang Zeng[1], Ziqi Rong [8], Tianmeng Chen [9], Timothy Billiar [9], Ying Ding [1], Heng Huang[6,7], Richard H. Duerr [3,10] ✉ & Wei Chen [1,5,10] ✉

Droplet-based single-cell sequencing techniques rely on the fundamental assumption that each droplet encapsulates a single cell, enabling individual cell omics profiling. However, the inevitable issue of multiplets, where two or more cells are encapsulated within a single droplet, can lead to spurious cell type annotations and obscure true biological findings. The issue of multiplets is exacerbated in single-cell multiomics settings, where integrating cross-modality information for clustering can inadvertently promote the aggregation of multiplet clusters and increase the risk of erroneous cell type annotations. Here, we propose a compound Poisson model-based framework for multiplet detection in single-cell multiomics data. Leveraging experimental cell hashing results as the ground truth for multiplet status, we conducted trimodal DOGMA-seq experiments and generated 17 benchmarking datasets from two tissues, involving a total of 280,123 droplets. We demonstrated that the proposed method is an essential tool for integrating cross-modality multiplet signals, effectively eliminating multiplet clusters in single-cell multiomics data—a task at which the benchmarked single-omics methods proved inadequate.

The rapid development of droplet-based single-cell sequencing methods has substantially improved biological insights into complex gene regulatory networks through various analyses, such as clustering analysis, differential expression analysis, and trajectory analysis. In droplet-based platforms, a common issue is that some droplets can encapsulate multiple cells rather than one or zero cells, leading to the formation of cell multiplets[1–3]. A nonignorable percentage of multiplets could become a key confounding factor for cell clustering or downstream analysis and lead to false biological discoveries. Therefore, multiplet identification and removal are important and fundamental steps in any single-cell data analysis workflow. Doublets and triplets are two major types of multiplets, with the former representing two cells captured together (the dominant case) and the latter representing three cells captured together (the less common case). While the terms "doublet" and "multiplet" are used interchangeably in most literature, our aim of explicitly modeling the multiplet formation

[1]Department of Biostatistics, University of Pittsburgh, Pittsburgh, PA 15213, USA. [2]Department of Epidemiology and Biostatistics, Memorial Sloan Kettering Cancer Center, New York, NY 10065, USA. [3]Department of Medicine, University of Pittsburgh, Pittsburgh, PA 15261, USA. [4]School of Medicine, Tsinghua University, 100084 Beijing, China. [5]Department of Pediatrics, University of Pittsburgh, Pittsburgh, PA 15224, USA. [6]Department of Computer Science, University of Maryland, College Park, MD 20742, USA. [7]Center of Bioinformatics and Computational Biology, College Park, MD 20740, USA. [8]School of Information, University of Michigan, Ann Arbor, MI 48109, USA. [9]Department of Surgery, University of Pittsburgh, Pittsburgh, PA 15213, USA. [10]Department of Human Genetics, University of Pittsburgh, Pittsburgh, PA 15261, USA. ✉e-mail: duerr@pitt.edu; wei.chen@pitt.edu

process necessitates differentiating them, and we specifically use "doublet" to refer to a droplet that contains exactly two cells.

Several experimental approaches have emerged for the detection of multiplets[1,4,5]. While these methods have demonstrated effectiveness in identifying and removing multiplets, their associated additional costs and labor requirements have resulted in rare use in practice. In contrast, computational methods that utilize the recorded data from each droplet to infer its multiplet status offers a favorable alternative that does not entail any additional data generation expenses. Computationally, a popular approach for multiplet detection using scRNA-seq data is to generate synthetic doublet labels through simulation (e.g., merging the expression profiles from two observed droplets to form an artificial doublet) and then classify the multiplets through semi-supervised learning[6–13]. Some more recent methods aim to build upon or integrate these existing methods to enhance the performance of multiplet detection[14–16]. Alternatively, several computational multiplet detection methods were tailored for leveraging single-cell assays for transposase-accessible chromatin sequencing (scATAC-seq) data, either by a similar semi-supervised learning approach[15,17] or by identifying chromatin regions with over two uniquely aligned reads[18]. The popular semi-supervised approach can effectively rank the droplets based on their similarities to simulated doublets. However, they typically use heuristic threshold selection for distinguishing between multiplets and singlets, which can cause an imbalance between precision and recall. This imbalance compromises multiplet detection reliability. In addition, the existing semi-supervised methods typically utilize the top highly variable genes (HVGs) as the input features, which makes them sensitive to heterotypic multiplets (i.e., multiplets formed by different types of cells) but not to homotypic multiplets (i.e., multiplets formed by the same type of cells).

Recent advances in single-cell experiment technologies have enabled simultaneous measurement of bimodal and trimodal single-cell data. Bimodal single-cell data typically include transcriptomic profiles combined with either cell surface protein data, as in CITE-seq and REAP-seq[19,20], or chromatin accessibility data, as in sci-CAR, SNARE-seq, and SHARE-seq[21–23]; common trimodal single-cell data are obtained by the simultaneous measurement of the transcriptome, cell surface protein, and chromatin accessibility (DOGMA-seq and TEA-seq)[24–26]. The integration of multiple modalities of information in single-cell analysis enhances the sensitivity of cell type identification but also increases susceptibility to the formation of multiplet clusters. Consequently, there is a heightened demand for tailored multiplet detection tools in single-cell multiomics data analysis. However, to the best of our knowledge, none of the widely available doublet/multiplet detection methods can effectively utilize single-cell multiomics data. While there have been efforts to utilize CITE-seq or VDJ-seq[27] data for multiplet detection, they demand specialized knowledge and subjective choices of thresholds from users, such as manual gating based on the co-expression of mutually exclusive surface protein markers[28].

In this study, we propose the COMpound POiSson multIplet deTEction (COMPOSITE) model, the first statistical model tailored for multiplet detection in single-cell multiomics data. COMPOSITE innovatively utilizes stable features, which are ideal for the multiplet detection problem since their recorded values are more closely related to multiplet status compared to highly variable features. Additionally, COMPOSITE conducts statistically rigorous inference on the probability of multiplets, thereby attaining an optimal balance between precision and recall for multiplet detection in practical settings. COMPOSITE further leverages a statistical approach to integrate multiomics information, which substantially enhances its multiplet detection performance. Due to the lack of public single-cell multiomics datasets with annotated multiplet status, we performed trimodal DOGMA-seq experiments with the cell hashing technique and generated 17 single-cell multiomics datasets in a total of 280,123 droplets with experimental ground truth of multiplet status. We also

demonstrated the generalizability of COMPOSITE by applying it to two additional datasets, each featuring cell types different from those present in the 17 datasets used for benchmarking. We illustrate the exceptional and robust performance of COMPOSITE in these single-cell multiomics datasets. We have implemented COMPOSITE into a Python package as well as a cloud-based application with a user-friendly interface.

## Results

### Compound Poisson framework for multiomics multiplet detection

Our proposed COMPOSITE method utilizes a statistical model to provide an automated framework for multiplet detection (Fig. 1a). To our knowledge, this is the first statistical multiplet detection model that is compatible with both single-omics and multiomics single-cell data. Specifically, our current model is compatible with three popular single-cell omics data modalities: scRNA-seq, antibody-derived tags (ADT, measuring surface protein epitopes), and scATAC-seq (measuring chromatin accessibility). In contrast to the prevailing single-cell data analysis methods that heavily depend on highly variable features[7,10,11], our proposed model harnesses the valuable information embedded in stable features[29]. Stable features exhibit minimal variability across different cells within a dataset, and the magnitude of their recorded values provides more accurate indications of multiplet status (Fig. S1a, b).

While multiplets generally exhibit higher stable feature values than singlets, individual stable features remain noisy with a broad range of values. This variability makes it challenging to infer the multiplet status based solely on individual stable feature values. To address this issue, COMPOSITE uses compound Poisson distributions to model the distributions of stable features. We make the following assumptions in the model: 1. For scRNA-seq and scATAC-seq modalities (with scATAC-seq data represented as gene activity inferred by Signac[30,31]), we assume that the contribution of a single cell to each recorded stable feature value within the droplet follows a gamma distribution (Fig. S1a, c). 2. For the ADT modality, we assume that the contribution of a single cell to each recorded stable feature value within the droplet follows a Gaussian distribution (Fig. S1d). 3. We assume that the recorded stable feature values in a multiplet depend on the summed contributions of each cell within the multiplet. 4. We model the number of cells present in each droplet using a Poisson distribution. These assumptions fully specify a compound Poisson-Gamma distribution for each stable feature in the RNA and ATAC modalities and a compound Poisson–Gaussian distribution for each stable feature in the ADT modality. Based on the compound Poisson distributions, within each modality, we combine all selected stable features across all cells to calculate the joint likelihood and estimate the parameter values through maximum-likelihood estimation. Afterwards, we perform statistical inference on the multiplet status.

In single-cell multiomics settings, once we have obtained the inference results from each modality, we combine these results across modalities by assigning droplet-specific modality weights. These weights are calculated using a combination of overall modality goodness-of-fit and droplet-specific data consistencies for each modality. In general, higher overall weights are assigned to the modalities that exhibit better fits. Then, for each droplet, we refine the overall modality weights to obtain droplet-specific modality weights based on the noisiness in each modality of the droplet. Specifically, for droplets with noisy data within a modality, their weights for that modality are adjusted downward and adjusted upward otherwise.

### Single-cell multimodal omics with cell hashing experiments generate data with ground truth multiplet status

To evaluate the performance of COMPOSITE, due to the lack of public single-cell multiomics datasets with annotated multiplet

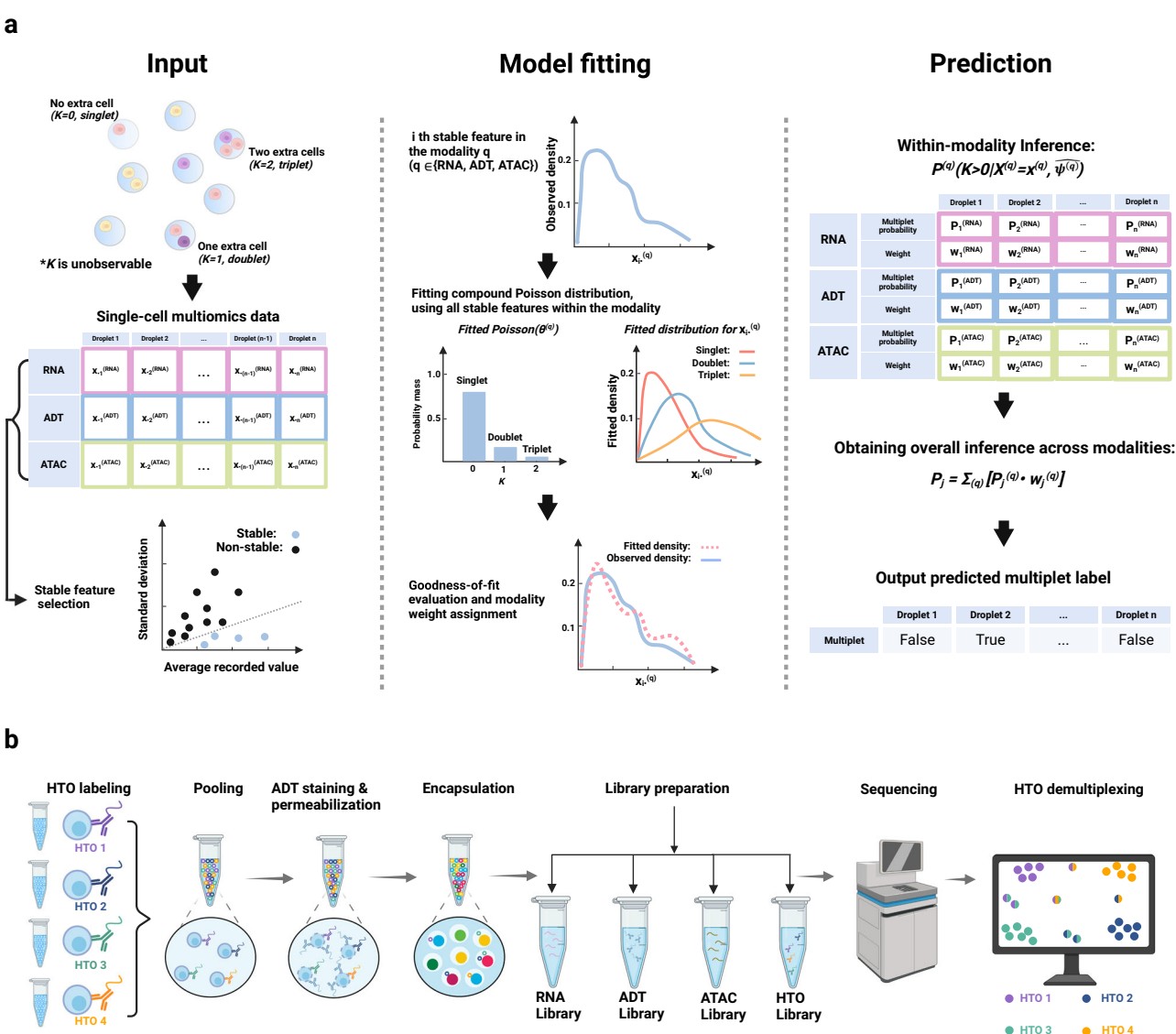

**Fig. 1 | Overview of the COMPOSITE model and the experimental workflow for generating the DOGMA-seq datasets with cell hashing-based ground truth.**
**a** The recorded data from a droplet can come from a single cell ($K = 0$), a composite of two cells ($K = 1$), or a composite of three cells ($K = 2$), where $K$ is an unobservable random variable representing the number of extra cells in the droplet. The COMPOSITE model can accept one or multiple modalities among RNA, ADT, and ATAC for each droplet as input, depending on data availability. The goal of the model is to infer $K$ for each droplet given the observed data. Stable features that display high mean-to-standard deviation values are selected for model fitting. In the fitted compound Poisson distribution, the Poisson component represents the estimated weights for each multiplet status, and the overall distribution of each stable feature breaks into a mixture of three conditional distributions with intrinsic summation relationships, respectively, for singlets, doublets, and triplets. The weight of each mixture component is characterized by the Poisson component. Then, the overall modality weights are assigned based on the goodness-of-fit of the corresponding modalities and are further adjusted for each droplet according to droplet-specific data quality for each modality. Afterwards, within each modality *(q)*, statistical inference on $K$ is performed based on the observed data *(x^(q))* and the estimated parameters *(ψ^(q))*. The final prediction is then the weighted combination of the predictions from each modality. **b** Workflow of the DOGMA-seq experiment with hashtag oligo (HTO) labeling. Before pooling the cells for the standard DOGMA-seq workflow, cells from different samples or different aliquots of the same sample were labeled with different HTOs. The HTO data from each droplet are used to infer the multiplet status, which is then regarded as the experimental ground truth multiplet label. We used eight HTOs in our experiments, but only four HTOs are illustrated in the figure due to space limitations. Figure created with BioRender.com, released under a Creative Commons Attribution-NonCommercial-NoDerivs 4.0 International license.

status, we conducted DOGMA-seq experiments to generate high-quality single-cell multiomics data with ground truth multiplet status obtained by cell hashing experiments (Fig. 1b). To generate single-cell multiomics datasets for comprehensive benchmarking, we conducted DOGMA-seq experiments using both T-cell-enriched peripheral blood samples and solid tissue samples (dissociated ileum mucosa biopsy immune-cell-enriched samples). We obtained ten peripheral blood T-cell-enriched and seven ileum immune-cell-enriched DOGMA-seq with cell hashing datasets involving a total of 280,123 droplets (Table S1). These

datasets represent the most extensive trimodal single-cell datasets with experimentally labeled multiplet status to date. To demonstrate the generalizability of our results, we incorporated two additional datasets, each generated independently and representing a wider variety of cell types. The first of these is derived from a colon biopsy sample, specifically enriched for non-immune cells (CD45−) through flow cytometry. Non-immune, epithelial, and mesenchymal cells in dissociated intestinal biopsies pose greater challenges for single-cell multiomics experiments. For this sample, we derived single-cell RNA data, which is

generally a more reliable data modality for non-immune cells. The second dataset, a PBMC DOGMA-seq dataset, was sourced from an independent study conducted at the University of Pittsburgh Medical Center. This contrasts with the 10 T cell-enriched peripheral blood samples produced by our laboratory, as it encompasses a more diverse array of blood cell types.

## COMPOSITE effectively captures the underlying mixture distribution of stable features

Since the RNA modality is the most common modality in most single-cell studies, we provide a detailed demonstration of the model fitting results on the RNA modality of an in-house peripheral blood dataset (PB-1). As an example, we visualized the ground truth distributions of one of the stable RNA features, *RPL11* (Fig. 2a, b). Without knowing the multiplet status of each droplet, the overall distribution of the *RPL11* expression level in all droplets is essentially a mixture distribution. However, the mixture distribution is dominated by singlets, and it is a challenging task for ordinary mixture models to capture the mixture components (Fig. 2a). In contrast, the COMPOSITE model handled this challenge well, and the gamma distributions inferred by the model closely matched the ground truth distributions (Fig. 2b, c). Additionally, the Poisson component of the COMPOSITE model also effectively captured the distribution of the number of cells within droplets (Fig. 2d). Hence, the COMPOSITE model can not only differentiate between multiplets and singlets but also provide an inference on the number of cells in the droplets. It calculates statistically meaningful probabilities associated with different multiplet statuses for each droplet. The most probable multiplet status is then the model classification result for that droplet. Importantly, the model classification results were close to the ground truth obtained by cell hashing (Fig. 2e).

COMPOSITE can also provide reliable prediction performance when applied to the ADT and scATAC-seq modalities. Notably, the goodness-of-fit metric we employed, the inverse of the Kolmogorov–Smirnov (KS) statistics[32–35], serves as a good indicator of model prediction performance (Fig. 2f). Therefore, in scenarios where only one modality of data is available, the goodness-of-fit (GOF) metric can assist in assessing the reliability of the model's prediction results. In general, a GOF value >3 indicates a good fit and reliable prediction performance.

## Multiomics data empowers COMPOSITE for enhanced multiplet detection

While COMPOSITE can provide reliable predictions when applied to a single modality, one of its main strengths lies in its ability to integrate information from multiple modalities, leading to enhanced performance. Its successful integration of information across modalities results from the sensible assignment of modality weights. COMPOSITE calculates droplet-specific modality weights based on the product of two components: 1. overall modality weights and 2. droplet-specific modality consistencies. The first component roughly determines the overall weight of each modality for all droplets, while the second component helps adjust the modality weights assigned to each droplet based on the noisiness in each modality of that droplet. To demonstrate the effectiveness of COMPOSITE in assigning droplet-specific modality weights, we visualized how these two components were related to the corresponding single modality prediction performance.

The first component, overall modality weights, is designed to be proportional to the overall goodness-of-fit of the corresponding modalities. The COMPOSITE model is a parametric statistical model, and its prediction performance relies on model fitting. Hence, the modalities with better model fitting are upweighted as they tend to

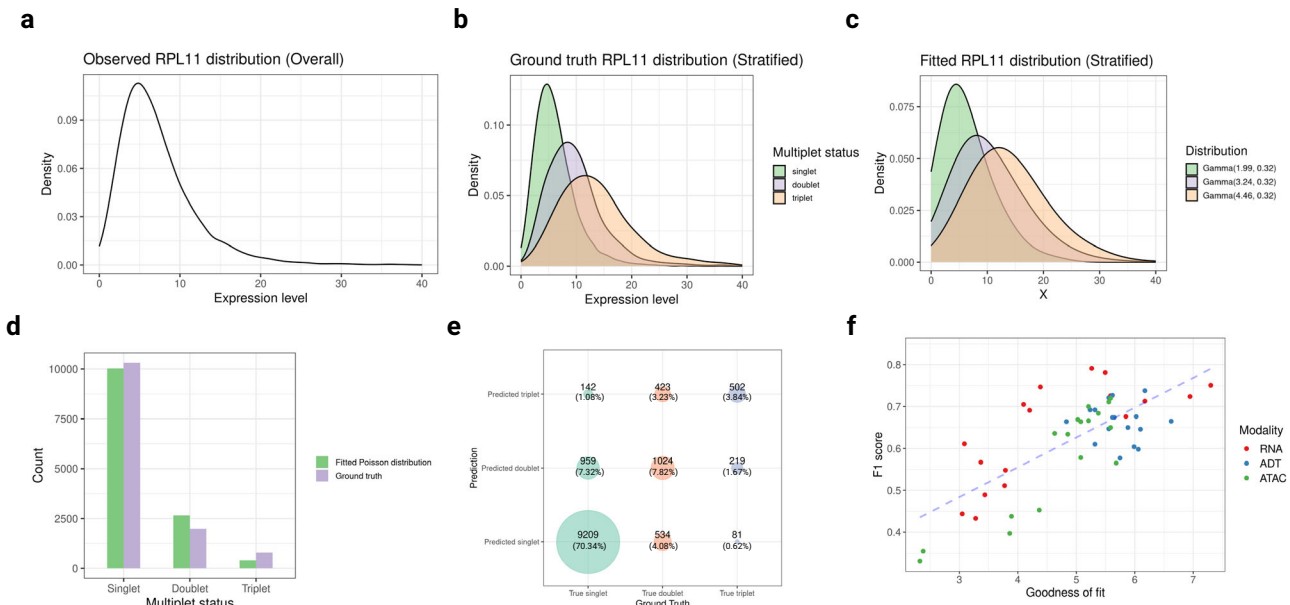

**Fig. 2 | COMPOSITE model fitting performance on single-omics data.** The data for **a**–**e** are from the RNA modality of the PB-1 dataset. **a**–**c** Using one of the stable RNA features (*RPL11*) to demonstrate how the compound Poisson-gamma distribution captures the ground truth distribution of the recorded expression level for different underlying multiplet statuses. **a** Observed overall distribution. **b** Observed distribution stratified by ground truth multiplet status. **c** Fitted Gamma distributions associated with each multiplet status. The parameters of the Gamma distributions were estimated by the compound Poisson-Gamma model. **d** Histogram comparing ground truth multiplet status distribution vs. multiplet status distribution simulated using Poisson(0.20), which is the fitted Poisson component for this dataset in the COMPOSITE framework. **e** Contingency table comparing ground truth vs. predicted multiplet status from COMPOSITE. The numbers on each intersection point of the grids represent the number and proportion of droplets that belong to the corresponding category, and the sizes of the dots on the grid intersections represent the magnitudes of the corresponding numbers. **f** Scatter plot displaying the relationship between the goodness-of-fit and the prediction performance in terms of F1 score. Each dot represents the prediction made using one modality of a specific dataset from the 17 in-house DOGMA-seq datasets. Source data are provided as a Source Data file.

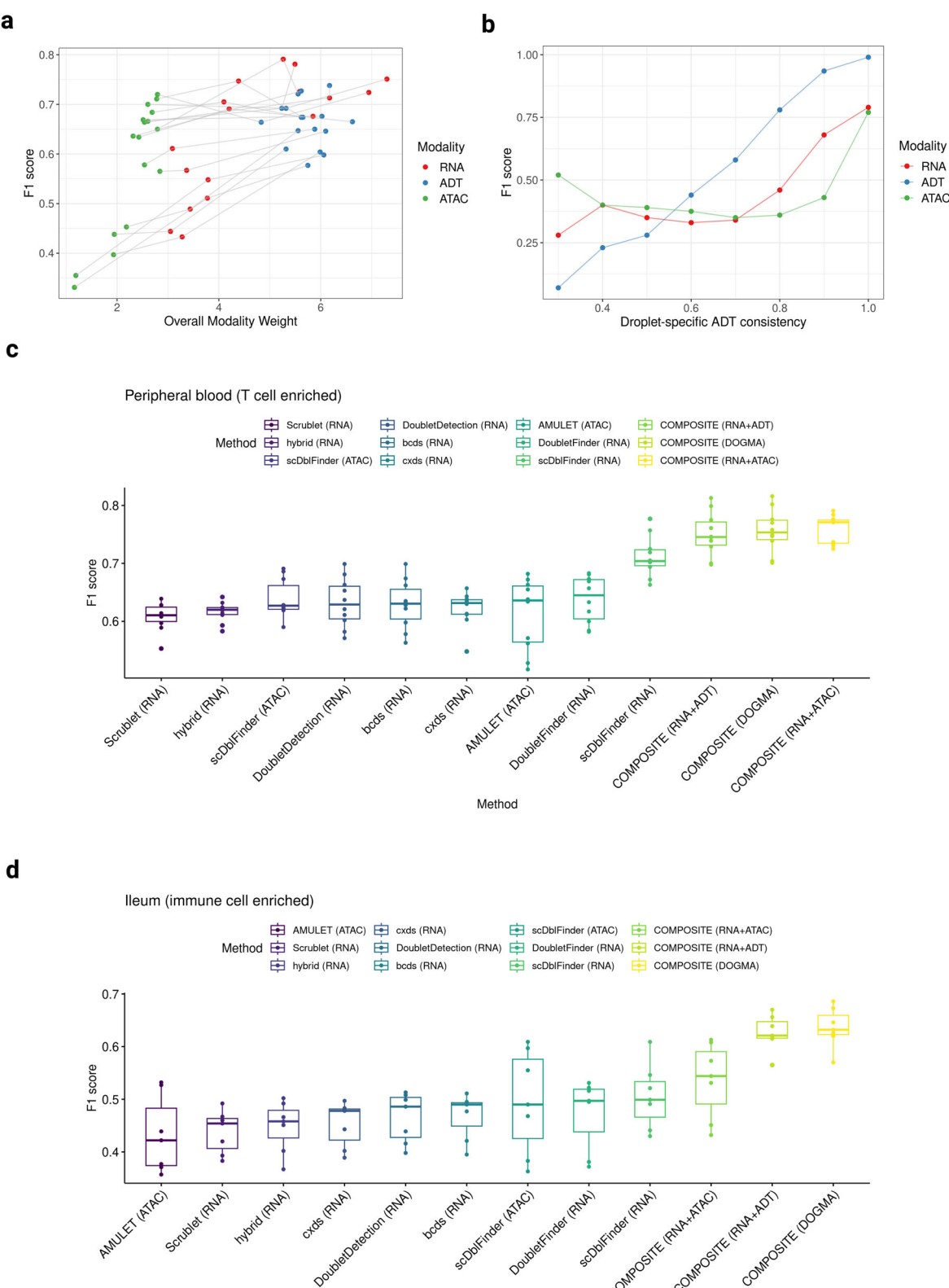

provide more reliable prediction results. To assess the usefulness of the overall modality weights, we visualized the relationship between single modality prediction performances (measured by F1 scores) and their corresponding overall modality weights for each of the 17 in-house DOGMA-seq samples (Fig. 3a). As expected, we noted that within each sample, the modalities that can provide better prediction performances were associated with higher overall modality weights in

general. These results indicate that the COMPOSITE model effectively upweights the modalities with better prediction performances.

The second component, droplet-specific modality consistencies, quantifies the level of consistency between the signals of individual stable features and the overall signal of the entire modality within a droplet. We visualized the relationship between the droplet-specific modality consistencies and the corresponding modality-specific

**Fig. 3 | COMPOSITE multiplet detection performance in single-cell multiomics setting. a** Scatter plot displaying the relationship between the overall modality weight and the prediction performance in terms of the F1 score. Gray lines connect the three modalities from the same dataset. The upward trends indicate that the modalities that can provide better prediction performances are associated with higher overall modality weights in general. This suggests that the COMPOSITE model effectively upweights the modalities with better prediction performances. **b** Prediction performances (in terms of F1 score) of each modality at different ADT consistency levels. The data are from the Ileum-1 dataset. **c** and **d** Boxplots showing the performances (in terms of F1 score) of each multiplet detection method on the in-house DOGMA-seq datasets ($n = 10$) from peripheral blood samples (**c**) and DOGMA-seq datasets ($n = 7$) from ileum biopsy samples (**d**). In the boxplots, the box spans from the first to the third quartile, with the median depicted as a line in the middle. The whiskers extend to 1.5 times the interquartile range (IQR). In the labels of the x-axis, the texts within the parenthesis indicate the modalities of data that were used as input into the corresponding method. Source data are provided as a Source Data file.

prediction performances within the Ileum-1 DOGMA-seq sample (Figs. 3b, S2a, b). We use the ADT modality as an example to illustrate the interpretation of these results. Specifically, we generated visualizations illustrating the prediction performance of each modality at different ADT consistency levels (Fig. 3b). To enhance the clarity of visualization, we rounded the ADT consistency of each droplet to the nearest decimal point. As the ADT consistency increases, the prediction performances based on the ADT modality demonstrate the most notable improvement compared to the RNA and ATAC modalities. These results indicate that COMPOSITE can effectively upweight the ADT modality for the droplets with good ADT modality prediction performances since the droplet-specific modality weights are defined to be proportional to droplet-specific modality consistencies. Similar results were observed when we stratified on RNA or ATAC consistency (Fig. S2a, b).

To demonstrate the enhanced performance of COMPOSITE when applied to single-cell multiomics data, we conducted a comparison of its single-omics and multiomics prediction results on our in-house DOGMA-seq datasets. Specifically, we considered various combinations of the available data to simulate different types of single-cell multiomics data generated from popular experimental techniques. These combinations include RNA + ATAC (simulating 10× multiome), RNA + ADT (simulating CITE-seq), and RNA + ADT + ATAC (DOGMA-seq). We use F1 score as the primary evaluation metric because it provides a balanced assessment of precision and recall, which makes it suitable for assessing the prediction performance in practical settings. For the peripheral blood samples, each combined multiomics prediction achieved a markedly higher median F1 score than any of the single-omics predictions within the combination (Fig. S3a; Table S2). Since the RNA and ADT modalities are generally assigned higher overall weights than the ATAC modality, the DOGMA-seq predictions are close to the RNA + ADT predictions. However, with the ATAC information added, every DOGMA-seq prediction still achieved observable improvement in the F1 score compared to the corresponding RNA + ADT prediction. For the ileum samples, multiomics combinations also achieved better performances in general, with the DOGMA-seq combination yielding the best predictions (Fig. S3b; Table S2). We noticed that the ATAC predictions for several of the ileum samples were unsatisfactory, with the F1 scores for three out of seven samples <0.4. However, when the ATAC data were combined with the RNA data, the combined predictions were greatly improved, with none of the F1 scores falling below 0.4. In addition to F1 scores, the area under the precision-recall curve (AUPRC) metric also indicates that multiomics prediction substantially improves the COMPOSITE performance over the single-omics prediction (Fig. S3c, d; Table S2).

**Multiplet removal with COMPOSITE effectively reduces bias in downstream analysis**
We evaluated the effects of using COMPOSITE for multiplet removal on downstream analyses, particularly focusing on clustering and trajectory inference, which serve as foundational steps for further analyses, such as differential expression (DE) analysis.

We use the PB-1 dataset as an example for illustration. In the clustering analysis, before multiplet removal, clusters 3−6 all have

significant proportions of multiplets (Fig. S4a, b), potentially biasing the downstream analysis. After multiplet removal with COMPOSITE (DOGMA), none of the identified clusters contain a significant amount of multiplets, thus reducing potential biases in subsequent analyses (Fig. S4c, d). For trajectory inference, from the PB-1 dataset, we extracted CD4+ T cells based on ADT expressions in scenarios both before and after multiplet removal with COMPOSITE (DOGMA) (Fig S5a, b). Subsequently, we performed trajectory inference independently for each scenario using Monocle 3[36] (Fig. S5c, d). Before the removal of multiplets, trajectory analysis revealed two branches, marked by red circles in Fig. S5c, extending into clusters identified as multiplets according to experimental ground truth (Fig. S5c, e). In contrast, after applying COMPOSITE (DOGMA) for multiplet removal, these aberrant branches were no longer present, indicating a cleaner trajectory inference free from the influence of multiplets (Fig. S5d, f).

**Comparison with existing multiplet detection methods**
One fundamental challenge faced by existing methods is the selection of an appropriate cut-off for distinguishing multiplets from singlets after ranking the droplets from the most likely multiplets to the least likely multiplets. Some of them select the classification threshold using a heuristic computational approach[7,12,15], while others estimate multiplet rates based on cell loading densities in the experiments[6,10,11]. The heuristic selection of thresholds lacks statistical rigor and may compromise the reliability of multiplet classification, and estimating multiplet rates based on cell loading densities is inherently unreliable, as the multiplet rate can vary significantly even when the cell loading density remains constant. Unlike the existing methods, COMPOSITE is a statistical model-based method that utilizes statistical inference to infer the multiplet status of each droplet, and it does not need a threshold selection process. Furthermore, the primary advantages and novelties of COMPOSITE lie in its capacity to leverage single-cell multiomics data for enhanced performance. Due to the lack of widely available multiomics multiplet detection methods, we benchmarked COMPOSITE with the state-of-the-art single-omics multiplet detection methods. For the scRNA-seq multiplet detection methods, we selected scDblFinder[15], DoubletFinder[10], DoubletDetection[12], scds[6] (including the bcds, cxds, and hybrid versions), and Scrublet[11] for benchmarking, as they had good performances in a previous benchmarking paper on single-omics multiplet detection[37,38]. For the scATAC-seq multiplet detection methods, we selected scDblFinder and AMULET, where scDblFinder is compatible with both RNA and ATAC data, and AMULET is the only existing read count-based multiplet detection method. scDblFinder and AMULET do not require manual parameter tuning, and we used the default parameter setting (selecting 0.05 as the false discovery rate cut-off for AMULET). The other methods require parameter tuning and manual selection of the expected multiplet rate. We estimated the multiplet rates for the methods that accept manual cut-off selections based on the cell loading densities following the guidelines provided by 10X Genomics (https://kb.10xgenomics.com/hc/en-us/articles/360054599512-What-is-the-cell-multiplet-rate-when-using-the-3-CellPlex-Kit-for-Cell-Multiplexing-). In terms of COMPOSITE, we considered the following three popular combinations: RNA + ATAC (simulating 10x multiome), RNA + ADT (simulating CITE-seq), and

RNA + ADT + ATAC (DOGMA-seq). We evaluated the performances of these methods in terms of their real-world application, where the ultimate goal for using multiplet detection methods is to predict the multiplet status of each droplet, rather than solely ranking the droplets based on their likelihood of being multiplets. Therefore, we used the F1 score as the evaluation metric because it considers the balance between precision and recall, which is crucial in practical settings. We compared their prediction performances on all 17 DOGMA-seq datasets. COMPOSITE consistently achieved markedly higher median F1 scores in both peripheral blood (Fig. 3c) and ileum biopsy samples (Fig. 3d), demonstrating enhanced multiplet detection capabilities and an optimal balance between precision and recall.

We provide a detailed illustration of the multiplet prediction outcomes using a peripheral blood dataset (PB-1) and an ileum biopsy dataset (Ileum-1). For COMPOSITE, we present its prediction results on the three modalities of DOGMA-seq, as this combination demonstrated the best performance and robustness (Fig. 3c, d), and we also present its prediction results on each single modality for a comparison with other single-omics approaches. For comparison, we visualized the prediction results from three existing methods, selected either for their unique approaches or for achieving high F1 scores in our benchmarking: scDblFinder, DoubletFinder, and AMULET.

For the PB-1 dataset, all these methods were successful in eliminating a significant proportion of ground truth multiplets and displayed their importance in practice (Figs. 4a–d and S6a–d). COMPOSITE (RNA) achieved comparable results to the other single-omics multiplet detection methods, while COMPOSITE (DOGMA) displayed the highest sensitivity for multiplet detection. The most prominent differences between COMPOSITE (DOGMA) and the other methods on the weighted nearest neighbors[39] (WNN) UMAP plots are in the circled cluster (Fig. 4a–d), where COMPOSITE (DOGMA) almost completely removed the circled cluster while the other methods, including the experimental cell hashing approach, only removed part of it. After removing the ground truth multiplets, the Azimuth[39] annotation results based on the ADT modality indicate that the circled cluster is unusual, as it was annotated as a mixture of B cells, CD4 T cells and CD8 T cells (Fig. 4e). By checking the quality control metrics (Fig. S7a–c), we confirmed that these cells are not low-quality singlets but are likely multiplets since they have unusually high total gene expression levels. Within the circled cluster, we also observed the co-expression of multiple exclusive ADT markers, including CD19 and CD3 (Fig. 4f–h), CD4 and CD8 (Fig. S8a–c), and CD4 and CD16 (Fig. S8d–f), which provide strong evidence that the circled cluster is a multiplet cluster. Therefore, in Fig. 4a, most of the droplets that appeared to be false positives are, in fact, true multiplets that were not identified by the experimental ground truth labeling. Despite being regarded as the gold standard for multiplet detection[40], the experimental cell hashing technique is unable to identify multiplets in droplets containing cells with identical hashtag oligos (HTOs). This limitation introduces minor inaccuracies in the ground truth labels, contributing to its inability to remove the circled multiplet cluster (Fig. 4e). The high sensitivity of COMPOSITE (DOGMA) to the multiplets in the circled cluster was contributed by the multiplet signals from the ADT modality. When using only the ADT data, COMPOSITE was still able to remove most of the droplets in the circled cluster (Fig. S6a). Importantly, none of those visualized exclusive ADT markers were selected as the stable ADT features for input into the COMPOSITE model, showcasing the model's robust performance independent from the highly variable features.

In comparison, within the same dataset (PB-1), we visualized the prediction results of COMPOSITE (DOGMA), COMPOSITE (RNA), scDblFinder (RNA), and DoubletFinder (RNA) on the UMAP plots generated using only the RNA modality (Fig. S9a–d). The multiplet clusters detected by these scRNA-seq multiplet detection methods closely aligned with the ground truth on the RNA UMAP plots, especially for COMPOSITE (RNA) and scDblFinder (RNA). After removing the ground truth multiplets, the droplets co-expressing exclusive ADT markers (Fig. S10a–k) or showing multiplet-like quality control metrics (Fig. S11a–c) did not concentrate in any specific clusters, indicating that the unremoved multiplets induced little bias to analytical outcomes. In contrast, in the previous single-cell multiomics settings, the capacity of WNN clustering to integrate information from multiple modalities for enhanced cell type identification simultaneously increased its susceptibility to biases from multiplet signals across modalities. This inadvertently led to the identification of a spurious cell cluster, even when experimental ground truth multiplets had been removed. Specifically, in the PB-1 dataset, if the circled WNN cluster remained unremoved (Fig. 4e), it might be mistakenly classified as CD3+/CD19+ B cells (Fig. 4f–h); this cell type is rare but has been recognized for its biological significance in prior single-cell studies[41,42]. Among all evaluated methods, including the experimental cell hashing technique, which served as ground truth, COMPOSITE (DOGMA) was the only one to successfully eliminate the circled cluster (Fig. 4a). These results highlight the importance of integrating multiplet signals across modalities for multiplet detection in single-cell multiomics settings and demonstrate the substantial value of COMPOSITE even when experimental ground truth data are available. The comparisons of these methods on the UMAP plots for the other nine peripheral blood samples are in Figs. S12–S20.

For the Ileum-1 dataset (Figs. 5a–d and S21a–d), the most obvious discrepancies between COMPOSITE (DOGMA) and the existing methods were highlighted by the red ellipses on the UMAP plots, where the droplets were mostly annotated as Natural Killer (NK) cells (Fig. 5e). The quality control metrics (Fig. S22a–c) and exclusive ADT markers (Fig. S23a–h) did not provide much evidence that these droplets were the multiplets missed by cell hashing. Therefore, we can assert with a high degree of confidence that within the regions marked by the ellipse, the single-modality multiplet detection methods—particularly those based on semi-supervised learning—have produced numerous false positives, leading to the erroneous exclusion of a substantial number of NK cells. These results indicate that the semi-supervised learning-based methods lack robustness in the smaller clusters. In contrast, in the highlighted areas, the false positive rate for COMPOSITE (DOGMA) remained low. In addition, compared to COMPOSITE (RNA) (Fig. 5b), COMPOSITE (DOGMA) (Fig. 5a) yielded significantly fewer false positive results in the circled cluster, indicating that COMPOSITE can effectively integrate information across modalities to minimize false detections of multiplets. The comparisons of these methods on the UMAP plots for the other six ileum samples are in Figs. S24–S29.

We evaluated the generalizability of COMPOSITE by testing it on two additional datasets, each representing a broad spectrum of cell types. The first dataset was derived from a colon biopsy sample enriched for non-immune cells (CD45−) via flow cytometry. Single-cell multiomics experiments on solid tissues face greater challenges than those on immune cells, particularly due to the less developed protocols for surface protein analysis, which typically focus on immune cell markers. We obtained scRNA-seq data for 6784 droplets from this challenging tissue type for benchmarking. As expected for non-immune, epithelial, and mesenchymal cells in dissociated intestinal biopsies, the data was relatively poor in quality in comparison to that from immune cells. In this context, the single-modality application of COMPOSITE still surpassed all other existing methods (Fig. S30). That demonstrates the effectiveness of COMPOSITE on relatively poor-quality datasets from challenging cell types.

The second dataset is a PBMC DOGMA-seq dataset, featuring DOGMA-seq data across 9643 droplets. This dataset was obtained

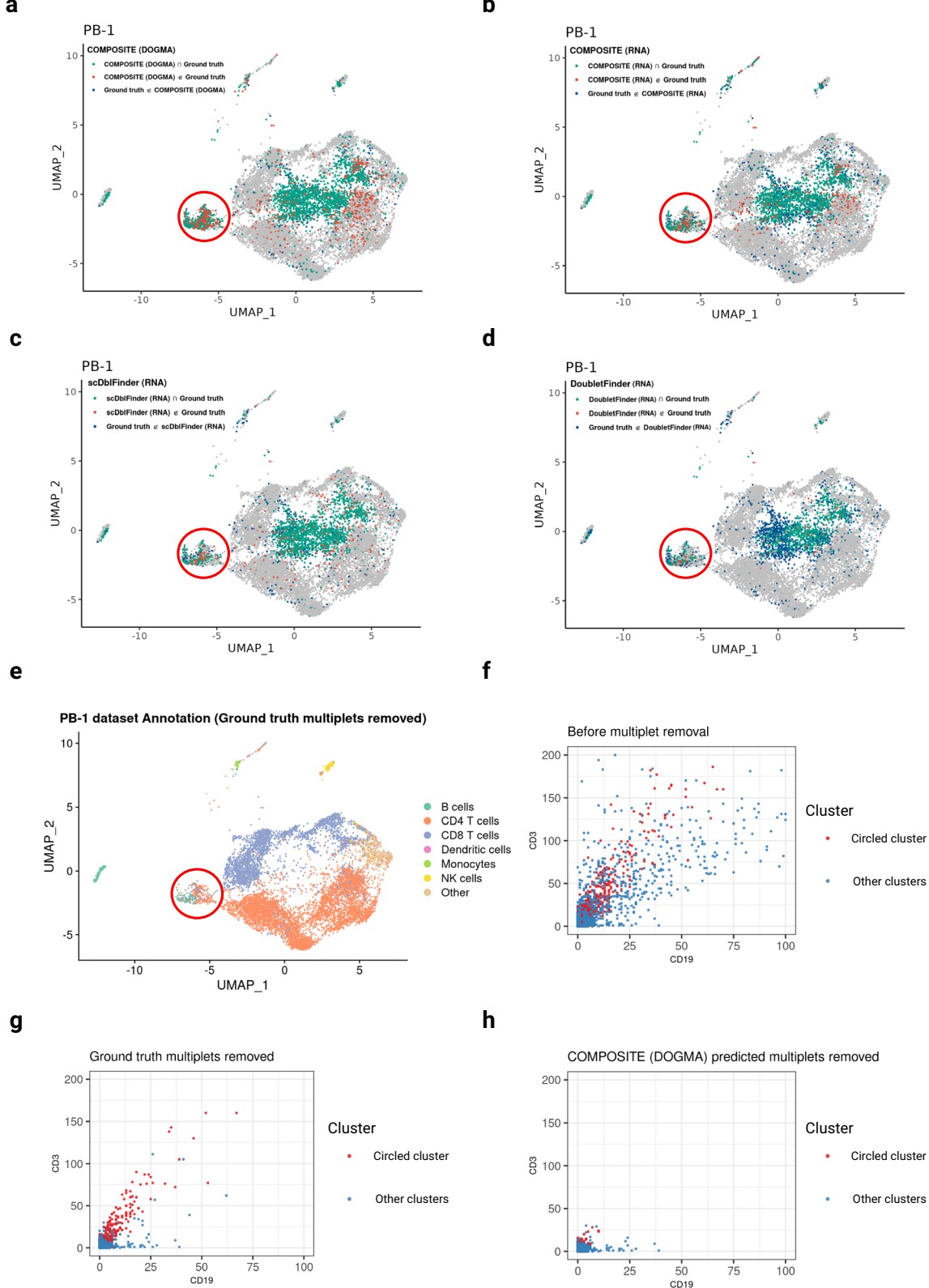

from an independent study conducted by a separate laboratory at the University of Pittsburgh Medical Center. In contrast to our laboratory's ten T cell-enriched peripheral blood samples, this dataset provided a more varied representation of blood cell types. Within this dataset, COMPOSITE demonstrated superior performance over existing methods, showing particular improvement when leveraging information from the ADT modality (Fig. S31).

## Simulation results demonstrate COMPOSITE's proficiency in handling homotypic and larger multiplets

Through simulation, we compared COMPOSITE with scDblFinder, the leading competitor based on real-data benchmarking, to highlight COMPOSITE's capabilities on (1) the identification of homotypic multiplets and (2) handling the datasets with a significant proportion of larger multiplets, such as triplets. For simulation, we first removed all

**Fig. 4 | Benchmarking of multiplet prediction methods on the PB-1 dataset. a–d** UMAP plots displaying the comparison between multiplet predictions and the ground truth on the PB-1 dataset. The multiplet detection methods shown are COMPOSITE (DOGMA) (**a**), COMPOSITE (RNA) (**b**), scDblFinder (RNA) (**c**), and DoubletFinder (RNA) (**d**). True positive (Prediction ∩ Ground truth), false positive (Prediction ∉ Ground truth), and false negative (Ground truth ∉ Prediction) predictions for multiplets are highlighted with green, red, and dark blue, respectively. Comparing the results from **a–d**, the circled cluster shows the most prominent difference among the prediction results from different methods. COMPOSITE (DOGMA) almost completely removed the circled cluster, while the other methods, including the experimental cell hashing approach, only removed part of it.

**e** Azimuth[39] cell type annotation based on the ADT data after removing ground truth multiplets. The clustering and UMAP visualization were generated from weighted nearest neighbors using all three modalities of data[39]. The red circle in **e** marks the cluster where the droplets were predicted to be various contradicting cell types by Azimuth. **f–h** Scatter plots comparing the expression of two exclusive ADT markers, CD3 and CD19, among all droplets before multiplet removal (**f**), among the remaining droplets after removing ground truth multiplets (**g**), and among the remaining droplets after removing COMPOSITE (DOGMA)-predicted multiplets (**h**). In **f–h**, the droplets that belong to the circled cluster in **a-e** are highlighted in red. Source data are provided as a Source Data file.

ground truth multiplets from the datasets and then simulated artificial multiplets by combining expression profiles from ground truth singlets within the datasets.

For homotypic multiplet simulation, we focused on CD4+ T cells and simulated 20 datasets that only contained artificial homotypic doublets simulated by combining expression profiles of two CD4+ T cells. In comparison, we simulated 20 general DOGMA-seq datasets with artificial doublet generated by aggregating expression profiles from pooled ground truth singlets from the peripheral blood samples. Specifically, for each scenario, two simulated datasets were generated from each of the 10 peripheral blood sample datasets. We then compared COMPOSITE (DOGMA) to scDblFinder (RNA) using the datasets simulated under these two settings. Compared to the general setting (Fig. S32a), COMPOSITE (DOGMA) displayed greater superiority over scDblFinder (RNA) in the homotypic setting (Fig. S32b). Therefore, the simulation results support COMPOSITE's effectiveness in detecting homotypic multiplets, surpassing existing methods that largely rely on highly variable genes.

To evaluate the performance of COMPOSITE on datasets with significant proportions of large multiplets, such as triplets, we conducted simulations across scenarios with artificial doublet rates from 5% to 30% and triplet rates from 2% to 8%, generating 20 simulated datasets for each scenario. Specifically, two simulated datasets were generated from each of the 10 peripheral blood sample datasets. Our results consistently show the superior performance of COMPOSITE (DOGMA) over scDblFinder (RNA), particularly as the rate of triplets increases (Fig. S33a–d). These results suggest that by distinguishing doublets and triplets in the modeling process, COMPOSITE can effectively handle the datasets with significant proportions of larger multiplets.

## Discussion

We have developed COMPOSITE, a unified statistical model based on the compound Poisson framework, which exhibits exceptional performance in multiplet detection, especially within the realm of single-cell multiomics. COMPOSITE can effectively leverage stable features within each modality to identify multiplet signals and combine these signals across modalities to enhance multiplet detection performance.

Notably, COMPOSITE is the first statistical model tailored to utilize single-cell multimodal information for multiplet detection. By employing statistical inference, COMPOSITE produces more reliable prediction results compared to existing methods in practical applications. Additionally, we demonstrated through simulation that the use of stable features endows COMPOSITE with robustness against both heterotypic and homotypic multiplets, whereas the existing methods only demonstrate sensitivity to heterotypic multiplets due to their reliance on highly variable features. One major limitation of COMPOSITE, like many other parametric statistical models, is that its performance can be negatively impacted when the assumptions are violated. When the data are too sparse, and the distributions of stable features cannot be adequately fitted by the model, the method's performance may lack robustness. To address this issue, we offer goodness-of-fit metrics that aid in evaluating the fitting of the model and inferring the reliability of its predictions.

We expect that COMPOSITE will offer substantial benefits to all laboratories engaged in single-cell experiments, particularly those aiming to conduct large-scale single-cell multiomics experiments. Single-cell multiomics data can reveal cell types and states that may not be distinguishable using a single modality of data. However, it also increases the likelihood of multiplet cluster isolation and the potential generation of spurious cell types. COMPOSITE harnesses multiplet signals across modalities to effectively identify and remove multiplets in multiomics settings, significantly enhancing the reliability of downstream analyses.

In addition, COMPOSITE aids in reducing cost and addressing the scalability issue of single-cell multiomics experiments. Single-cell multiomics profiling provides great potential for a deeper understanding of the relationship among different modalities. However, single-cell multiomics experiments, such as DOGMA-seq and CITE-seq, can be financially demanding, and the cost factor is a major bottleneck for their widespread adoption. Increasing the cell loading densities in such experiments can effectively reduce the average library prep cost per cell. However, this increase in throughput also yields a higher rate of multiplets, compromising data reliability and resulting in inaccurate biological discoveries. While experimental approaches exist for multiplet detection and removal, they incur additional costs. COMPOSITE, on the other hand, provides a rapid and robust solution for multiplet removal without any additional expenses, effectively reducing costs by enabling high throughput while preserving the reliability of biological inferences.

COMPOSITE is currently compatible with three modalities of data (scRNA-seq, ADT, and scATAC-seq). However, it offers a flexible framework that can be expanded to accommodate additional modalities as new techniques and data types emerge. Moreover, while the COMPOSITE model is specifically designed for multiplet detection, its compound Poisson statistical framework has wide applicability in the modeling of single-cell data. For example, for spatial transcriptomic data, the compound Poisson framework is potentially helpful in estimating the number of cells and unraveling the cell composition within each spot[43–45]. We anticipate that these interesting directions will be explored in future studies.

By leveraging GPU acceleration, the COMPOSITE method can efficiently perform multiplet detection for a single 10X Chromium well within minutes (Fig. S34). Multiple-well data will be processed in parallel, so our streamlined approach should be applied to any droplet-based experiments without concern of computational burden.

By applying COMPOSITE to DOGMA-seq datasets from both blood samples and solid tissue samples, we showcased its outstanding and consistent efficacy in multiplet detection. The COMPOSITE pipeline is available as a cloud-based application with a user-friendly interface https://shiny.crc.pitt.edu/shinyproj_composite/. We anticipate COMPOSITE to be an invaluable tool for enhancing the reliability of biological inferences and achieving cost reduction in all single-cell studies. It is particularly beneficial in single-cell multiomics studies, not

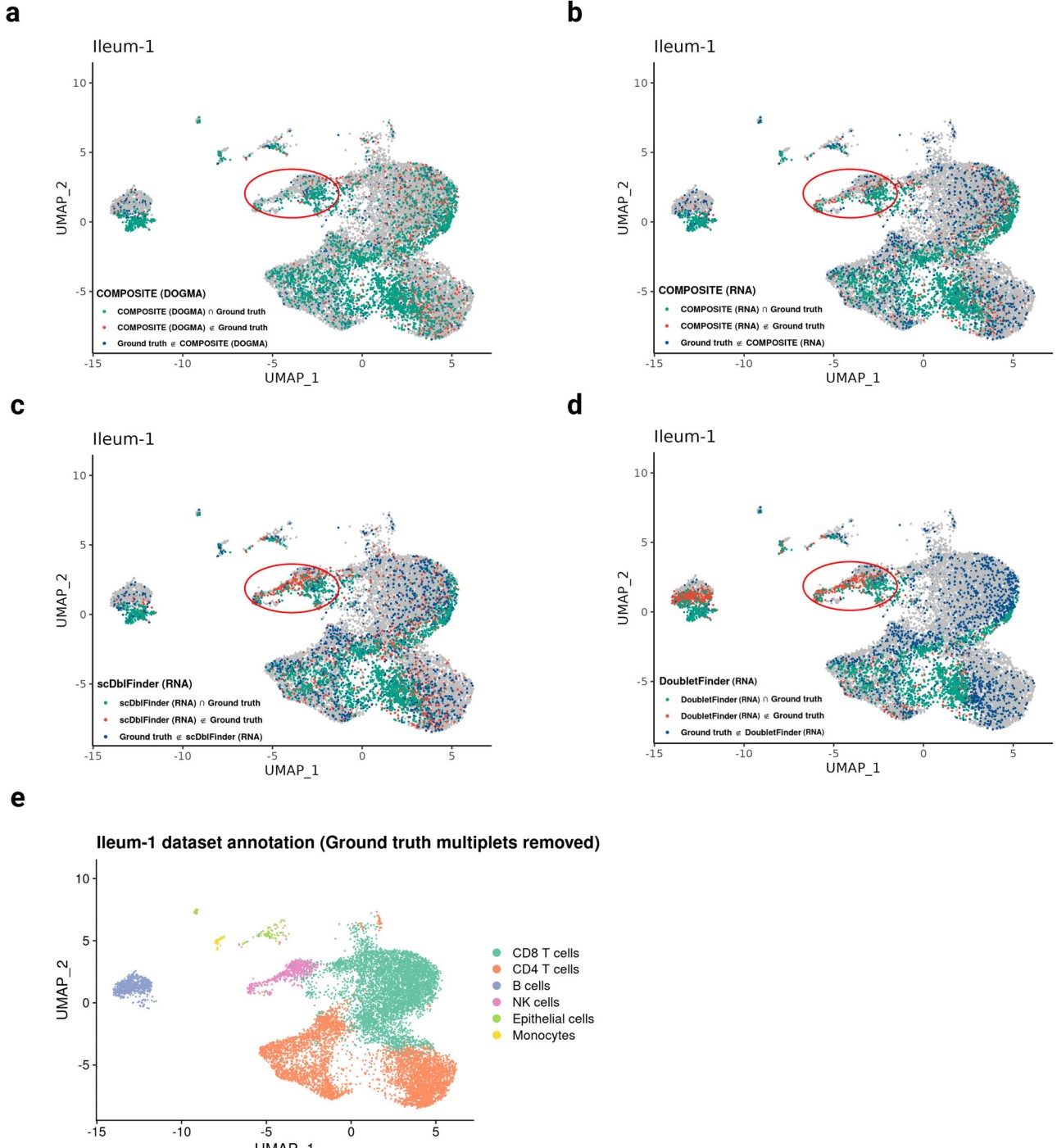

**Fig. 5 | Benchmarking of multiplet detection on the Ileum-1 dataset. a–d** UMAP plots displaying the comparison between multiplet predictions and the ground truth on the Ileum-1 dataset. The multiplet detection methods shown are COMPOSITE (DOGMA) (**a**), COMPOSITE (RNA) (**b**), scDblFinder (RNA) (**c**), and DoubletFinder (RNA) (**d**). True positive (Prediction ∩ Ground truth), false positive (Prediction ∉ Ground truth), and false negative (Ground truth ∉ Prediction) predictions for multiplets are highlighted with green, red, and dark blue, respectively. Comparing the results from **a–d**, the circled cluster shows the most prominent difference among the prediction results from different methods. In the circled cluster, the three prediction methods based on RNA data resulted in many false positives (**b–d**), while the false positive rate for COMPOSITE (DOGMA) remained low (**a**). **e** Manual cell type annotation based on ADT data after removing ground truth multiplets. The clustering and UMAP visualization were generated from weighted nearest neighbors using all three modalities of data[39]. Source data are provided as a Source Data file.

only due to its exceptional performance for multiomics multiplet detection but also because the expensive single-cell multiomics experiments are in greater demand for multiplet detection tools due to the generally more significant multiplet issues resulting from high-density cell loading in cost-saving efforts.

## Methods

### The COMPOSITE model

The COMPOSITE model is designed to perform multiplet detection for both single-omics and multiomics single-cell data. It follows a four-step approach: (1) stable feature selection, (2) multiplet detection using

single-omics data, (3) model fitting evaluation, and (4) integrating multiomics data if available.

## Stable feature selection

In contrast to most existing multiplet detection methods that primarily rely on highly variable features, the COMPOSITE model takes a different approach by leveraging stable features. Because they remain relatively stable across different cell types in the dataset, their abundance is more closely related to multiplet status (Fig. S1a−d). Different datasets may have different stable features; hence, we require a stable feature selection process for every dataset. For RNA and ADT modalities, the stable feature selection is the only pre-processing step required for COMPOSITE, and we have implemented that in the COMPOSITE pipeline. For the ATAC modality, the Signac[30,31] pipeline needs to be applied to infer the gene activity matrix to reduce sparsity, and then the stable features are selected from the inferred gene activity matrix. To select stable features, we adapt the analytic framework in scMerge[46]. In this paper, stable features were used for dataset integration. Although the goals are different, the stable feature selection process is adaptable to our problem. The key principle is to select features that have a low proportion of zero values among droplets and show a high signal-to-noise ratio. We first compute the proportion of zero counts among all droplets for each feature and then screen out features with a proportion of zero values higher than 50%. The second step is to compute the mean/standard deviation (SD) ratio for each feature that passes the first criterion, and features with the highest mean/SD ratios are considered candidate features for multiplet detection (Fig. S35). It is worth noting that the data are $\log 1p$ transformed ($\log 1p(x) = \log(x + 1)$) when computing the mean and SD to avoid the influence of outlier expression. In practice, the model performance increases with the number of stable features selected at first and then reaches a plateau (Fig. S36). In our study, we used 300 stable features for RNA and ATAC modalities. For the ADT modality, we use the top 10% most stable features instead of setting a fixed number, as the number of ADT features may vary in different experiments. Users may adjust this threshold to include more or fewer stable features, and the COMPOSITE model is robust to variations in the number of stable features.

## Multiplet detection using single-omics data

After stable feature selection, the COMPOSITE model takes the selected stable features as input. In this section, we describe the COMPOSITE prediction on single-omics data.

To specify the statistical model, we define the following notations:

$q \in \{$RNA,ADT,ATAC$\}$ denotes the indicator of modalities.

$X_{ij}^{(q)}$ denotes the random variable that models the value of the $i$th stable feature ($i = 1, \ldots, m^{(q)}$) of the $j$th droplet ($j = 1, \ldots, n$) in modality $q$.

$K_j$ denotes the random variable that models the number of extra cells in the $j$th droplet, $K_j = 0, 1, \ldots, \infty$. $K_j = 0$ indicates that the $j$th droplet is a singlet. $K_j + 1$ is the total number of cells in the $j$th droplet.

$Z_{ijl}^{(q)}$ denotes the random variable that models the contribution of the $j$th ($l = 0, 1, \ldots, K_j$) extra cell in the $j$th droplet to $X_{ij}^{(q)}$, where $l = 0$ indicates the first cell in the droplet. Note that there is no intrinsic order for the cells in a droplet and we define the order only for model specification. The relationship between $X_{ij}^{(q)}$ and $Z_{ijl}^{(q)}$ is as follows:

$$X_{ij}^{(q)} = \sum_{l=0}^{K_j} Z_{ijl}^{(q)} \qquad (1)$$

Our goal is to perform statistical inference on the number of extra cells in each droplet given the observed data for that

droplet. That is, we need to calculate $P(K_j | \mathbf{X_j^{(q)}} = \mathbf{x_j^{(q)}})$. Then, $1 - P(K_j = 0 | \mathbf{X_j^{(q)}} = \mathbf{x_j^{(q)}})$ is the probability that the $j$th droplet is a multiplet. Here, the lower-case characters, such as $\mathbf{x}$, denote the specific values of the corresponding random variables.

Although the single-cell features are recorded as count data, the stable features have high recorded values in general and their distributions are close to continuous distributions. Based on the observed distribution of the data (Fig. S1a−d), we make the following assumptions on the distributions of the random variables:

1. If $q = $ RNA or ATAC, $Z_{ijl}^{(q)}$ follows a Gamma distribution.
2. If $q = $ ADT, $Z_{ijl}^{(q)}$ follows a Gaussian distribution.
3. $Z_{ijl}^{(q)}$'s (for $l = 0, 1, \ldots, K_j$) are independent and identically distributed (i.i.d), conditional on $K_j$.
4. $K_j \sim$ Poisson$(\theta^{(q)})$, where $\theta^{(q)}$ is shared across all droplets, i.e., all droplets are assumed to have the same probabilities for capturing extra cells.

The parameters for characterizing the distribution of $Z_{ijl}^{(q)}$ depend on $K_j$, the number of extra cells in the droplet. When the number of cells within a droplet increases, the expected contributions of individual cells within that droplet will decline (Fig. S37). We define this effect as the "decline effect" and introduce a decline parameter $d^{(q)}$ to model this effect. The expected feature values for multiplets containing $k + 1$ cells can be characterized by

$$E\left(X_{ij}^{(q)} | K_j = k\right) = (k+1)^* f\left(d^{(q)}, k\right) ^* E\left(Z_{ij0}^{(q)} | K_j = 0\right) \qquad (2)$$

$$E\left(X_{ij}^{(q)} | K_j = k\right) = (k+1)^* f\left(d^{(q)}, k\right) ^* E\left(X_{ij}^{(q)} | K_j = 0\right) \qquad (3)$$

where $k > 0$ and $f\left(d^{(q)}, k\right) < 1$. We assume that in a multiplet with $k + 1$ cells ($k > 0$), the expected feature values are greater than those in singlets but are less than $k + 1$ times those values in singlets. We express this assumption using the following set of inequalities:

$$\begin{cases} E\left(X_{ij}^{(q)} | K_j = k\right) > E\left(X_{ij}^{(q)} | K_j = 0\right) \\ E\left(X_{ij}^{(q)} | K_j = k\right) < (k+1)^* E\left(X_{ij}^{(q)} | K_j = 0\right) \end{cases}, \text{for } k > 0 \qquad (4)$$

We define $f\left(d^{(q)}, k\right)$ to have the following form such that $E\left(X_{ij}^{(q)}, | K_j = k\right)$, as expressed in Eq. (3), satisfies the set of inequalities (4):

$$f\left(d^{(q)}, k\right) = \left[ \frac{1}{k+1} + \left(1 - \frac{1}{k+1}\right) \frac{1}{1 + \exp\left(-d^{(q)}\right)} \right] \qquad (5)$$

where $d^{(q)} \in (-\infty, +\infty)$ is the decline parameter shared by all droplets within the same modality in a dataset. Then, we can fully specify the distributions of $Z_{ijl}^{(q)}$ and $X_{ij}^{(q)}$.

For $q = $ RNA or ATAC,

$$Z_{ijl}^{(q)} \sim \text{Gamma}\left(\left[\frac{1}{K_j+1} + \left(1 - \frac{1}{K_j+1}\right) \frac{1}{1 + \exp\left(-d^{(q)}\right)}\right] \alpha_i^{(q)}, \beta_i^{(q)}\right) \qquad (6)$$

which ensures that the set of inequalities (4) is satisfied. The distribution of $X_{ij}^{(q)}$ is then a compound Poisson-Gamma distribution. The probability density function (PDF) of $X_{ij}^{(q)}$ is:

$$f_{X_{ij}^{(q)}}(x) = \sum_{k=0}^{\infty} P\left(K_j = k\right) f_{X_{ij}^{(q)}}\left(x | K_j = k\right) \qquad (7)$$

$$f_{X_{ij}^{(q)}}(x) = \sum_{k=0}^{\infty} P\left(K_j = k\right) f_{\sum_{l=0}^{k} Z_{ijl}^{(q)}}(x) \tag{8}$$

where $P\left(K_j = k\right)$ is the Poisson probability mass function (PMF):

$$P\left(K_j = k\right) = \frac{\theta^{(q)k} \exp\left(-\theta^{(q)}\right)}{k!} \tag{9}$$

Note that we also considered and implemented zero-inflated Poisson (ZIP). The fitting results typically assign close-to-zero weights to the zero component, causing the distribution to degenerate into an ordinary Poisson distribution. Therefore, here we do not separately introduce the ZIP case. Since $Z_{ijl}^{(q)}$'s ($l = 0, \ldots, K_j$) are i.i.d random variables, by the properties of Gamma distribution,

$$\left(\sum_{l=0}^{K_j} Z_{ijl}^{(q)} | K_j = k\right) \sim \text{Gamma}\left(\left[\frac{1}{k+1} + \left(1 - \frac{1}{k+1}\right)\frac{1}{1 + \exp\left(-d^{(q)}\right)}\right](k+1)\alpha_i^{(q)}, \beta_i^{(q)}\right) \tag{10}$$

In theory, $k$ can go to infinity, but in our implementation, considering the practical needs and computational efficiency, we do not model beyond triplets in the default setting ($k_{\max} = 2$) but allow users to change it if desired. After setting a $k_{\max}$ value, the Poisson distribution is right truncated. The right truncated Poisson distribution PMF is

$$P_{\text{T}}\left(K_j = k\right) = \frac{\frac{\theta^{(q)k} \exp\left(-\theta^{(q)}\right)}{k!}}{\sum_{p=0}^{k_{\max}} \frac{\theta^{(q)p} \exp\left(-\theta^{(q)}\right)}{p!}}, k \in \{0, 1, \ldots, k_{\max}\} \tag{11}$$

We assume that the droplets are independent, then the joint PDF for stable feature $i$ across all droplets is

$$f_{\boldsymbol{X}_{i\cdot}^{(q)}}(x) = \prod_{j=1}^{n} \sum_{k=0}^{k_{\max}} P_{\text{T}}\left(K_j = k\right) f_{\sum_{l=0}^{k} Z_{ijl}^{(q)}}(x) \tag{12}$$

Note that most of the stable features, such as housekeeping genes and mitochondrial genes, do not have regulatory relationships with each other and are biologically independent. Moreover, conditional on the same multiplet status, the correlations among stable features are close to zero (Fig. S1e, f). Therefore, we assume that the stable features are independent conditional on the number of cells in the droplet. The joint PDF for all droplets across all stable features within modality $q$ is then:

$$f_{\boldsymbol{X}^{(q)}}(x) = \prod_{i=1}^{m^{(q)}} \prod_{j=1}^{n} \sum_{k=0}^{k_{\max}} P_{\text{T}}\left(K_j = k\right) f_{\sum_{l=0}^{k} Z_{ijl}^{(q)}}(x) \tag{13}$$

We use $\boldsymbol{\psi}^{(\boldsymbol{q})}$ to denote the set of parameters that need to be estimated, i.e.,

$$\boldsymbol{\psi}^{(\boldsymbol{q})} = \left\{\theta^{(q)}, \alpha_i^{(q)}(i = 1, \ldots, m^{(q)}), \beta_i^{(q)}(i = 1, \ldots, m^{(q)}), d^{(q)}\right\} \tag{14}$$

The joint log-likelihood is

$$l\left(\boldsymbol{\psi}^{(\boldsymbol{q})} | \boldsymbol{X}^{(q)} = \boldsymbol{x}^{(q)}\right) = \log \prod_{i=1}^{m^{(q)}} \prod_{j=1}^{n} \sum_{k=0}^{k_{\max}} P_{\text{T}}\left(K_j = k\right) f_{\sum_{l=0}^{k} Z_{ijl}^{(q)}}(x) \tag{15}$$

$$l\left(\boldsymbol{\psi}^{(\boldsymbol{q})} | \boldsymbol{X}^{(q)} = \boldsymbol{x}^{(q)}\right) = \sum_{j=1}^{n} \sum_{i=1}^{m^{(q)}} \log\left(\sum_{k=0}^{k_{\max}} P_{\text{T}}\left(K_j = k\right) f_{\sum_{l=0}^{k} Z_{ijl}^{(q)}}(x)\right) \tag{16}$$

Within each modality, the parameter values are estimated through maximum likelihood maximization. Once the parameter values $\widehat{\boldsymbol{\psi}^{(\boldsymbol{q})}}$ have been estimated, the next step is to perform statistical inference on $K_j$'s to predict the number of cells in each droplet:

$$P\left(K_j = k | X_j^{(q)} = \boldsymbol{x}_j^{(q)}, \widehat{\boldsymbol{\psi}^{(\boldsymbol{q})}}\right) = \frac{P_{\text{T}}\left(K_j = k, \boldsymbol{x}_j^{(q)} | \widehat{\boldsymbol{\psi}^{(\boldsymbol{q})}}\right)}{P_{\text{T}}\left(\boldsymbol{x}_j^{(q)} | \widehat{\boldsymbol{\psi}^{(\boldsymbol{q})}}\right)} \tag{17}$$

$$P\left(K_j = k | X_j^{(q)} = \boldsymbol{x}_j^{(q)}, \widehat{\boldsymbol{\psi}^{(\boldsymbol{q})}}\right) = \frac{f_{\boldsymbol{X}_j^{(q)}}\left(\boldsymbol{x}_j^{(q)} | K_j = k, \widehat{\boldsymbol{\psi}^{(\boldsymbol{q})}}\right) P_{\text{T}}\left(K_j = k | \widehat{\boldsymbol{\psi}^{(\boldsymbol{q})}}\right)}{\sum_s f_{\boldsymbol{X}_j^{(q)}}\left(\boldsymbol{x}_j^{(\boldsymbol{q})} | K_j = s, \widehat{\boldsymbol{\psi}^{(\boldsymbol{q})}}\right) P_{\text{T}}\left(K_j = s | \widehat{\boldsymbol{\psi}^{(\boldsymbol{q})}}\right)} \tag{18}$$

$$P\left(K_j = k | \boldsymbol{X}_j^{(q)} = \boldsymbol{x}_j^{(q)}, \widehat{\boldsymbol{\psi}^{(\boldsymbol{q})}}\right) = \frac{\prod_{i=1}^{m^{(q)}} \left[f_{X_{ij}^{(q)}}\left(x_{ij}^{(q)} | K_j = k, \widehat{\boldsymbol{\psi}^{(\boldsymbol{q})}}\right)\right] P_{\text{T}}\left(K_j = k | \widehat{\boldsymbol{\psi}^{(\boldsymbol{q})}}\right)}{\sum_s \prod_{i=1}^{m^{(q)}} \left[f_{X_{ij}^{(q)}}\left(x_{ij}^{(q)} | K_j = s, \widehat{\boldsymbol{\psi}^{(\boldsymbol{q})}}\right)\right] P_{\text{T}}\left(K_j = s | \widehat{\boldsymbol{\psi}^{(\boldsymbol{q})}}\right)} \tag{19}$$

where the distribution of $(X_{ij}^{(q)} | K_j = \mathrm{k}, \widehat{\boldsymbol{\psi}^{(\boldsymbol{q})}})$ is fully specified as

$$\text{Gamma}\left(\left[\frac{1}{k+1} + \left(1 - \frac{1}{k+1}\right)\frac{1}{1 + \exp\left(-\widehat{d^{(q)}}\right)}\right](k+1)\widehat{\alpha_i^{(q)}}, \widehat{\beta_i^{(q)}}\right) \tag{20}$$

The probability that the $j$ th droplet is a singlet is then $P\left(K_j = 0 | \boldsymbol{X}_{\cdot j}^{(q)} = \boldsymbol{x}_j^{(q)}, \widehat{\boldsymbol{\psi}^{(\boldsymbol{q})}}\right)$. We classify the $j$ th droplet as a singlet if that probability is ≥0.5 and classify it as a multiplet otherwise.

For $q = \text{ADT}$,

$$Z_{ijl}^{(q)} \sim \text{Gaussian}\left(\left[\frac{1}{K_j + 1} + \left(1 - \frac{1}{K_j + 1}\right)\frac{1}{1 + \exp\left(-d^{(q)}\right)}\right]\mu_i^{(q)}, \sigma_i^{2(q)}\right) \tag{21}$$

By default, we still assume that the stable features are independent conditional on the number of cells in the droplet, and this default setting can handle most of the cases. Nonetheless, for ADT data, it is possible that the selected stable features are not biologically independent, because in some experiments, there are too few ADT features, and all of them are highly variable across different cell types. Therefore, for the ADT modality we also implemented a model assuming non-zero correlation between the features:

$$\boldsymbol{Z}_{jl}^{(\boldsymbol{q})} \sim \text{MVG}\left(\left[\frac{1}{K_j + 1} + \left(1 - \frac{1}{K_j + 1}\right)\frac{1}{1 + \exp\left(-d^{(q)}\right)}\right]\boldsymbol{\mu}^{(q)}, \boldsymbol{\Sigma}^{(q)}\right) \tag{22}$$

The set of parameters that need to be estimated for the ADT modality is

$$\boldsymbol{\psi}^{(\boldsymbol{q})} = \left\{\theta^{(q)}, \boldsymbol{\mu}^{(q)}, \boldsymbol{\Sigma}^{(q)}, d^{(q)}\right\}, q = \text{ADT} \tag{23}$$

By default, for $q = \text{ADT}$, the expression of the joint likelihood is the same as *(16)*, and the only difference is that the distribution of $\sum_{l=0}^{k} Z_{ijl}^{(q)}$ becomes:

$$\left(\sum_{l=0}^{K_j} Z_{ijl}^{(q)} | K_j = k\right) \sim \text{Gaussian}$$
$$\left(\left[\frac{1}{k+1} + \left(1 - \frac{1}{k+1}\right)\frac{1}{1 + \exp\left(-d^{(q)}\right)}\right](k+1)\mu_i^{(q)}, k\sigma^{2(q)}_i\right) \quad (24)$$

The statistical inference on $K_j$'s follows the same derivations as in Eqs. (17)−(19):

$$P\left(K_j = k | \boldsymbol{X}_j^{(q)} = \boldsymbol{x}_j^{(q)}, \widehat{\boldsymbol{\psi}^{(q)}}\right) = \frac{\prod_{i=1}^{m^{(q)}}\left[f_{X_{ij}^{(q)}}\left(x_{ij}^{(q)} | K_j = k, \widehat{\boldsymbol{\psi}^{(q)}}\right)\right]P_{\text{T}}\left(K_j = k | \widehat{\boldsymbol{\psi}^{(q)}}\right)}{\sum_s \prod_{i=1}^{m^{(q)}}\left[f_{X_{ij}^{(q)}}\left(x_{ij}^{(q)} | K_j = s, \widehat{\boldsymbol{\psi}^{(q)}}\right)\right]P_{\text{T}}\left(K_j = s | \widehat{\boldsymbol{\psi}^{(q)}}\right)}$$
$$(25)$$

The only difference from the scenarios where $q = \text{RNA}$ or ATAC is that the distribution of $(X_{ij}^{(q)} | K_j = k, \widehat{\boldsymbol{\psi}^{(q)}})$ is given by:

$$\text{Gaussian}\left(\left[\frac{1}{k+1} + \left(1 - \frac{1}{k+1}\right)\frac{1}{1 + \exp\left(-\widehat{d^{(q)}}\right)}\right](k+1)\widehat{\mu_i^{(q)}}, k\widehat{\sigma^{2(q)}_i}\right)$$
$$(26)$$

The probability that the $j$ th droplet is a singlet is then $P\left(K_j = 0, | \boldsymbol{X}_{\cdot j}^{(q)} = \boldsymbol{x}_{\cdot j}^{(q)}, \widehat{\boldsymbol{\psi}^{(q)}}\right)$ calculated according to Eq. (25).

## Model fitting evaluation

Since COMPOSITE is a parametric statistical model, its performance is contingent upon how well it fits the data. We measure the goodness-of-fit based on the Kolmogorov–Smirnov (KS) statistics. The KS statistic measures the maximum difference between the empirical cumulative distribution function (CDF) and the fitted CDF. The empirical CDF for the $i$ th stable feature in the modality $q$ is defined as:

$$F_{i,(q)}^{(n)}(x) = \frac{1}{n}\sum_{j=1}^{n} I(x_{ij}^{(q)} \leq x) \quad (27)$$

The fitted CDF for the $i$ th stable feature in the modality $q$ is:

$$\hat{F}_{i,(q)}(x) = P(X_{ij}^{(q)} \leq x | \widehat{\boldsymbol{\psi}^{(q)}}) = \sum_s P\left(K_j = s\right) P(X_{ij}^{(q)} \leq x | K_j = s, \widehat{\boldsymbol{\psi}^{(q)}}), \forall j \in \{1, \ldots, n\} \quad (28)$$

The KS statistic for the $i$ th stable feature in modality $q$ is then:

$$\text{KS}_i^{(q)} = \max_x \left|F_{i,(q)}^{(n)}(x) - \hat{F}_{i,(q)}(x)\right|, x \in \mathbb{R}_+ \quad (29)$$

We take the average of the KS statistics across all stable features within modality $q$ to get the average KS statistic for modality $q$:

$$\overline{\text{KS}^{(q)}} = \frac{1}{m^{(q)}}\sum_{i=1}^{m^{(q)}} \text{KS}_i^{(q)} = \frac{1}{m^{(q)}}\sum_{i=1}^{m^{(q)}} \max_x \left|F_{i,(q)}^{(n)}(x) - \hat{F}_{i,(q)}(x)\right|, x \in \mathbb{R}_+ \quad (30)$$

We define the overall goodness-of-fit for modality $q$ as

$$\text{GOF}^{(q)} = \frac{1}{\overline{\text{KS}^{(q)}}} \quad (31)$$

This goodness-of-fit metric is a good indicator of the reliability of the prediction result. A higher goodness-of-fit value is associated with better model prediction performance.

## Integrating multiomics data

If multi-modality (multiomics) data are available, COMPOSITE first obtains the prediction from each individual modality of data and then combines the prediction across modalities by assigning droplet-specific modality weights. COMPOSITE calculates droplet-specific modality weights based on the product of two components: 1. overall modality weights, and 2. droplet-specific modality consistencies.

For the first component, the basic idea is that the modalities with better goodness-of-fit values are anticipated to demonstrate superior prediction performance and should be assigned higher overall weights. Therefore, the overall weight for each modality is defined to be proportional to the overall goodness-of-fit of the modality. The overall modality weight for modality $q$ is denoted as $W^{(q)}$:

$$W^{\text{RNA}} = \text{GOF}^{\text{RNA}} = \frac{1}{\overline{\text{KS}}^{\text{RNA}}} \quad (32)$$

$$W^{\text{ADT}} = \text{GOF}^{\text{ADT}} = \frac{1}{\overline{\text{KS}}^{\text{ADT}}} \quad (33)$$

$$W^{\text{ATAC}} = \lambda \cdot \text{GOF}^{\text{ATAC}} = \lambda\frac{1}{\overline{\text{KS}}^{\text{ATAC}}}, \lambda \in [0,1] \quad (34)$$

We down weight the ATAC modality since predictions based on ATAC data generally lack sensitivity. By default, we set $\lambda = 0.5$.

The second component, droplet-specific modality consistencies, measures how consistent the signals from individual stable features are with the overall signal from the entire modality within a droplet. To calculate the droplet-specific modality consistencies, we compare the posterior probability of singlet conditional respectively on the single stable feature values and all stable feature values within the modality. The posterior probability of a singlet for the $j$ th droplet given the observed value of a single stable feature $i$ in the $q$ th modality is:

$$P\left(K_j = 0 | x_{ij}^{(q)}, \widehat{\boldsymbol{\psi}^{(q)}}\right) = \frac{f\left(x_{ij}^{(q)} | K_j = 0, \widehat{\boldsymbol{\psi}^{(q)}}\right)P\left(K_j = 0 | \widehat{\boldsymbol{\psi}^{(q)}}\right)}{\sum_s f\left(x_{ij}^{(q)} | K_j = s, \widehat{\boldsymbol{\psi}^{(q)}}\right)P\left(K_j = s | \widehat{\boldsymbol{\psi}^{(q)}}\right)} \quad (35)$$

The posterior probability of a singlet for the $j$ th droplet given the observed value of all stable features in the $q$ th modality is:

$$P\left(K_j = 0 | \boldsymbol{x}_j^{(q)}, \widehat{\boldsymbol{\psi}^{(q)}}\right) = \frac{f\left(\boldsymbol{x}_j^{(q)} | K_j = 0, \widehat{\boldsymbol{\psi}^{(q)}}\right)P(K_j = 0 | \widehat{\boldsymbol{\psi}^{(q)}})}{\sum_s f\left(\boldsymbol{x}_j^{(q)} | K_j = s, \widehat{\boldsymbol{\psi}^{(q)}}\right)P(K_j = s | \widehat{\boldsymbol{\psi}^{(q)}})} \quad (36)$$

The droplet-specific modality consistency for the $j$ th droplet is

$$R_j^{(q)} = \sum_{i=1}^{m^{(q)}} \frac{I(u(P(K_j = 0 | x_{ij}^{(q)}, \widehat{\boldsymbol{\psi}^{(q)}})) = u(P(K_j = 0 | \boldsymbol{x}_j^{(q)}, \widehat{\boldsymbol{\psi}^{(q)}})))}{m^{(q)}} \quad (37)$$

where $I$ is the indicator function and

$$u(x) = \begin{cases} 0, \text{if } x > \frac{1}{2} \\ 1, \text{if } x \leq \frac{1}{2} \end{cases} \quad (38)$$

To define the cell-specific modality weight, we take the product of the two components and then pass it through the SoftMax

transformation to make it a proper weight:

$$w_j^{(q)} = \text{SoftMax}\left(W^{(q)} \times R_j^{(q)}\right) = \frac{e^{W^{(q)} \times R_j^{(q)}}}{\sum_q e^{W^{(q)} \times R_j^{(q)}}} \tag{39}$$

The final inference on $K_j$ is obtained by calculating a weighted sum of the inference results for $K_j$ from each individual modality:

$$P(K_j = k | \boldsymbol{x_j}, \widehat{\boldsymbol{\psi}}) = \sum_q w_j^{(q)} \times P\left(K_j = k | \boldsymbol{x_j^{(q)}}, \widehat{\boldsymbol{\psi^{(q)}}}\right) \tag{40}$$

To obtain the final multiomics prediction of the multiplet status of the $j$ th droplet, we compare $P(K_j = 0 | \mathbf{x}_j, \widehat{\boldsymbol{\psi}})$ to 0.5. We classify the $j$ th droplet as a singlet if that probability is ≥0.5 and classify it as a multiplet otherwise.

## DOGMA-seq experiment details

**Preparation of peripheral blood T cell DOGMA-seq datasets.** T cells were enriched from 15 ml of whole blood donated by young adult human study subjects using a magnetic bead-based negative selection method and were then cryopreserved. For each experiment, cells from one to four subjects were thawed simultaneously, and each subject's cells were cultured in different stimulation conditions, including Human CD3/CD28/CD2 T Cell Activator, IL-1B and IL-23, with or without Prostaglandin E2 (PGE2), for varying time periods. Following centrifugation and resuspension, ~300,000 cells from each condition were incubated with Human TruStain FcX™ (BioLegend) for 10 min on ice and then stained with a unique TotalSeq™-A anti-human Hashtag antibody (BioLegend) in 50 μl for 30 min at 4 °C. Cells were then washed in the Laminar Wash™ Mini System (Curiox Biosystems). After cell collection, up to 1.5 million cells from the 8 to 13 hashtagged samples were pooled. The cells were then stained with TotalSeq™-A Human Universal Cocktail, V1.0 (BioLegend), consisting of antibodies bound to antibody-derived tag (ADT) oligonucleotides. Cells were washed as described above. Finally, the cells underwent the CG000338 Chromium Next GEM Multiome ATAC plus Gene Expression Rev. D (10X Genomics) protocol for Gene Expression and ATAC library construction, to which adjustments were made following the "Cell permeabilization with Digitonin (DIG)" section of the DOGMA-seq (NYGC Innovation Lab) protocol[24] to accommodate the DOGMA-seq technique.

## Preparation of PBMC DOGMA-seq dataset

Blood samples were obtained from human study subjects. PBMCs were isolated by standard Ficoll centrifugation. The isolated cells were cryopreserved. This dataset comprises cells derived from four distinct samples, each originating from a unique subject. The four samples were processed in parallel, which were thawed in the 37 °C water batch and transferred to a 50 ml conical tube after thawing was complete. One milliliter of thawing medium (RPMI with 10% FBS) was added dropwise (5 s/drop), followed by 2, 4, 8, and 16 ml thawing medium at ~1-min intervals. The cell preparation steps for DOGMA-seq mirrored those described in the subsection "Preparation of peripheral blood T cell DOGMA-seq datasets." Generally, each sample was first split into two 1.5 ml low-binding tubes (~0.5 million cells/tube). Each tube was incubated with Human TruStain FcX™ (BioLegend) for 10 min and then incubated with a unique TotalSeq™-A anti-human Hashtag antibody (BioLegend) for 30 min. After 3 times of wash (1350 rpm*5 min, PBS + 2% FBS), 187,500 cells/tube from 8 tubes were pooled into one 1.5 ml low-binding tube and incubated with TotalSeq™-A Human Universal Cocktail, V1.0 (BioLegend) for 30 min. After 3 times of wash (1350 rpm*5 min, PBS + 2%FBS), cells were permeabilized with 100 μl Digitonin lysis buffer for 1 min and then washed with 1 mL Digitonin

wash buffer (pipette 5 times up and down, then 1350 rpm*5 min). All the reactions were performed at 4 °C. Finally, 30,000 nuclei were loaded on one well of 10x Genomics Chip. The DOGMA-seq library preparation process followed the same steps outlined in the subsection titled "Preparation of peripheral blood T cell DOGMA-seq datasets."

## Preparation for Crohn's disease ileum mucosa immune cells

For each Crohn's disease patient, biopsies from macroscopically non-inflamed and/or inflamed mucosa were obtained and then cryopreserved. For each experiment, six cryopreserved ileum mucosa samples from three patients were thawed, digested, and dissociated into a single-cell suspension, filtered through a 30-μm strainer, centrifuged, and resuspended. Cells from each mucosa were then split into two aliquots, and each of the 12 aliquots was stained with a unique Total-Seq™-A anti-human Hashtag antibody (BioLegend) plus Apotracker in 100 μl for 30 min at 4 °C. After filtering through a FACS tube strainer, 7AAD was added, and live, non-apoptotic cells (7AAD⁻ Apotacker⁻) in each aliquot were enriched by flow cytometry cell sorting. The vast majority of the sorted cells were immune cells, as most non-immune cells were in the excluded dead or apoptotic cell populations. The 12 post-sorted samples were centrifuged, resuspended, counted, and combined for a 10-min incubation with Human TruStain FcX™ (BioLegend) on the ice, followed by antibody staining with TotalSeq™-A Human Universal Cocktail, V1.0 (BioLegend) and TotalSeq™-A0123 anti-human CD326 (Ep-CAM) (BioLegend). Without washing, the cells underwent the CG000338 Chromium Next GEM Multiome ATAC plus Gene Expression Rev. D (10X Genomics) protocol for Gene Expression and ATAC library construction, which was adjusted following the "Cell permeabilization with PFA/LLL" section of the DOGMA-seq (NYGC Innovation Lab) protocol[24] to accommodate the DOGMA-seq technique.

## Preparation for Crohn's disease colon mucosa non-immune cells

Similar to the steps described above, four cryopreserved colon mucosa samples from two patients were processed into a single-cell suspension. Cells from each mucosa were then split into either one or two aliquots. After incubated with Human TruStain FcX™ (BioLegend), each of the aliquots was stained with a unique TotalSeq™-A anti-human Hashtag antibody (BioLegend), as well as anti-CD45 fluorescent antibody and Apotracker, and then washed. After cells were filtered through Flowmi strainer, 7AAD was added, and live, non-apoptotic, CD45⁻ cells (7AAD⁻ Apotacker⁻ CD45⁻) in each aliquot were enriched by flow cytometry cell sorting. The vast majority of the sorted cells were non-immune cells.

## DOGMA-seq library sequencing and preprocessing

The DOGMA-seq libraries were pooled, sometimes with libraries from other projects, and sequenced on the NovaSeq 6000 sequencing platform (Illumina) at the University of Pittsburgh Medical Center Genome Center. Cell Ranger ARC software (version 2.0.1) was used for processing and aligning sequenced RNA and ATAC libraries to the GRCh38 human reference genome, and Kallisto|Bustools workflow (kallisto version 0.46.1, bustools version 0.39.3) was used for aligning sequenced ADT and HTO libraries. After data filtering and modality combination (RNA + ATAC + ADT + HTO), HTO libraries were demultiplexed using GMMDemux[47]. The following groups of datasets originated from Gel Bead-In-EMulsion (GEM) wells loaded with distinct aliquots from the same pool of hashtagged cells: PB-1 and PB-2; PB-3 and PB-4; PB-5, PB-6, and PB-7; PB-8 and PB-9; Ileum-1 and Ileum-2; Ileum-3 and Ileum-4; Ileum-5 and Ileum-6.

## Model performance evaluation and benchmarking

In our study, two metrics were used for the evaluation of model performance: the area under the precision-recall curve (AUPRC) and the

*F*1 score. The *F*1 score is defined as

$$F1 \text{ score} = \frac{2*\text{precision}*\text{recall}}{\text{precision} + \text{recall}} \quad (41)$$

The AUPRC metric has been widely used for the benchmarking of multiplet detection methods[37], as it is suitable for the scenario where the two binary classes are imbalanced[48]. However, AUPRC is not sufficient for assessing the model performances in real applications. A model with a high AUPRC can still exhibit poor prediction performance if the classification threshold or cutoff is not appropriately chosen, particularly in cases of imbalanced precision and recall. The *F*1 score, however, provides a balanced assessment of precision and recall. Therefore, the *F*1 score provides a more comprehensive evaluation of the model performance in real applications, and we used the *F*1 score as the major evaluation metric for multiplet prediction performance.

### Reporting summary

Further information on research design is available in the Nature Portfolio Reporting Summary linked to this article.

## Data availability

All relevant data supporting the key findings of this study are available within the article and its Supplementary Information file. The cell hashing data and the single-cell stable feature data generated in this study have been deposited in Zenodo under accession code https://doi.org/10.5281/zenodo.11167173 [https://doi.org/10.5281/zenodo.11167174][49]. Source data are provided with this paper.

## Code availability

COMPOSITE has been implemented in the "sccomposite" Python package, which is available at https://github.com/CHPGenetics/COMPOSITE. Relevant experiment codes and results are available in the GitHub repository and on Zenodo under the accession code https://doi.org/10.5281/zenodo.11166717 [https://zenodo.org/records/11166718][50]. The COMPOSITE pipeline is also available as a cloud-based application https://shiny.crc.pitt.edu/shinyproj_composite/.

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

## Acknowledgements

This study is supported by NSF2225775 to W.C. and HHuang, U19AG055373 to W.C., U01DK062420 to R.D. and W.C., P01AI106684 to W.C., R35GM127027 to T.B. and W.C., and R01DK138458 to R.D. and W.C. R.D. and W.C. are also supported by a grant from The Leona M. and Harry B. Helmsley Charitable Trust to the University of Pittsburgh. X.W. is supported by the Memorial Sloan Kettering Cancer Center (P30 CA008748). This research is also supported in part by the University of Pittsburgh Center for Research Computing through the resources provided. Specifically, this work used the HTC cluster, which is supported by NIH award number S10OD028483. Figure 1, created with BioRender.com, was released under a Creative Commons Attribution-NonCommercial-NoDerivs 4.0 International license.

## Author contributions

H. Hu, X.W., R.D., and W.C. designed the research; W.C. and R.D. supervised the research; H. Hu, X.W., S.F., Z.X., J.L., E.H., Z.R., and T.C. performed the research; Y.C., M.Y., L.Z., and H. Huang provided knowledge for algorithm implementation; S.F., Z.X., E.H., T.C., T.B., and R.D. conducted biological experiments and provided guidance from a biological perspective; and H. Hu, X.W., S.F., Y.C., Y.D., and W.C. wrote the paper.

## Competing interests

The authors declare no competing interests.
