## [Peer Review File · Nature Communications]

A unified model-based framework for doublet or multiplet detection in single-cell multiomics dataReviewer #1 (Remarks to the Author):

Summary:

This manuscript presents a new method for single-cell multiplet detection specifically designed for multiomic data, which is an untapped area in the field of bioinformatics. COMPOSITE is able to make more accurate predictions of multiplet status for each droplet within a multimodal dataset compared to existing methods by using 1) stable (i.e., housekeeping) gene expression as features of importance, which they demonstrate as having more distinct distributions between singlets and multiplets when compared to highly variable genes, and 2) weighing the contribution of multiplet calls from each modality on a droplet-by-droplet basis based on consistency, which reduces the effect of noise on multiplet detection. The authors demonstrate the higher performance of their tool compared to existing methods on 17 immune datasets. Overall, I think this method is innovative, statistically-backed, and its good performance is well evidenced on the datasets presented in the manuscript. However, additional consideration should be given to the breadth of existing tools compared against, the use of ATAC-seq-based gene expression inference, and some claims of performance (homotypic doublets and insensitivity to the number of stable features) that are not supported in the data presented.

Major Concerns:

- The authors state that:

"The vast majority of the sorted cells were immune cells, as most non-immune cells were in the excluded dead or apoptotic cell populations." (lines 656-658)

This means that, in combination with the peripheral blood samples, that effectively all the datasets tested were in immune cells. However, we desperately need methods that can work on more tissues with cell types that may be less defined than those in blood. Perhaps using a cryopreservation technique or sample preparation method would allow for the retention of non-immune cells and would give users confidence that this method would work in their tissues. It is not uncommon for bioinformatics methods to work well in PBMCs and fail on other tissue types, so I think this should be addressed.

- Do you think that the lower goodness of fit and F1 scores for ATAC-seq data may be due to the use of inferred gene expression as the model input instead of some way that would allow you to use the raw data itself? The authors describe these data:

"with scATAC-seq data represented as gene activity inferred by Signac". (line 134)

The authors of Signac acknowledge the limitations of their website in the section describing how to perform this inference: "Note that the activities will be much noisier than scRNA-seq measurements. This is because they represent measurements from sparse chromatin data, and because they assume a general correspondence between gene body/promoter accessibility and gene expression which may not always be the case." In the vignette quoted above (https://stuartlab.org/signac/articles/pbmc_vignette), the authors of Signac are using the gene expression inference as a method for calling cell type, not to use it as bonafide expression data.

- The introduction and discussion both mention that this method is sensitive to homotypic doublets while the existing methods are not, but I don't see where this has been demonstrated in the manuscript. Specifically in the discussion the authors state:

"Additionally, the use of stable features endows COMPOSITE with robustness against both heterotypic and homotypic multiplets, whereas many existing methods only demonstrate sensitivity to heterotypic multiplets due to their reliance on highly variable features." (lines 374-376).

As far as I can tell, the authors do not demonstrate that their tool works to remove homotypic doublets, only stating that it theoretically should based on how the method works. Additionally, they state that when using COMPOSITE, multiplets:

"did not concentrate in any specific clusters, indicating that the unremoved multiplets induced little bias to analytical outcomes." (lines 332-333)

If these remaining multiplets are homotypic, as they should be given their location in the clusters, then it undermines the claims in the discussion.

Minor Concerns:

- I would have liked to see a wider range of other multiplet detection tools tested in addition to the three described in the manuscript, even if it was just a base level comparison of the performance on RNA-seq data alone to get a comparable F1 score.

- I would like to see the tutorial include the preprocessing steps for scanpy (Python) users, not just Seurat (R) users.
- The authors claim:

“the COMPOSITE model is robust to variations in the number of stable features” (lines 459-460). It would be good to demonstrate this in the manuscript.
- The description of the method itself is excellent and very detailed. It might be helpful to define lowercase “x” in lines 479 and 480 for those who are less familiar with the random variable vs. specific value notation.
- The authors state that:

“For the PB-1 dataset, all benchmarked methods were successful in eliminating a significant proportion of ground truth multiplets and displayed their importance in practice (Figures 4A-4D; Supplementary Figures 4B-4C)” (lines 297-299)

On the PB-1 dataset, we see UMAP visualizations for the following:

 - o COMPOSITE – RNA (Fig 4)
 - o scDblFinder – RNA (Fig 4)
 - o DoubletFinder – RNA (Fig 4)
 - o COMPOSITE – ADT (Sup. Fig 4)
 - o COMPOSITE – DOGMA (Fig 4)
 - o scDblFinder – ATAC (Sup. Fig 4)
 - o AMULET – ATAC (Sup. Fig 4)

Given this thoroughness for all methods and combinations of data, I would like to see COMPOSITE performance on the ATAC seq data alone on a UMAP in Supplementary Figure 4.
- I also want to note that Supplemental Figure 4B is called out before 4A in the text, which was a bit confusing. It might be worth checking the manuscript for other ordering issues such as this for reading clarity.

Reviewer #2 (Remarks to the Author):

In this work, the authors proposed a statistical model COMPOSITE for multiplet detection in single-cell multiomics data. COMPOSITE exhibited remarkable superiority over existing single-omics multiplet detection methods, based on the 17 trimodal DOGMA-seq experimental datasets. The authors demonstrate that COMPOSITE is an essential tool for integrating cross-modality multiplet signals. Specifically, I have some major comments as below.

- 1) In the results section, COMPOSITE has been applied to 17 datasets profiled from T-cell-enriched peripheral blood samples and solid tissue samples. However, the benchmarking of multiplet prediction methods is only specifically shown for PB-1 (Figures 2 and 4) and Ileum-1 dataset (Figure 5). What are the benchmarking results on the other datasets?
- 2) In Figure 2E, the contingency table shows the ground truth vs. predicted multiplet status from COMPOSITE. From this table, COMPOSITE tends to detect more doublets and triplets than ground truth. And Figure 2F shows the F1 scores on most datasets are below 0.7. I am quite concerned about the performance of COMPOSITE in detecting false positives of multipliers.
- 3) As shown in Figure 3C-D, in the peripheral blood samples, COMPOSITE shows quite better performance than existing methods. In contrast, the benchmarking performance on those methods are tend to be comparable on the ileum samples. I am questioning whether the performance of COMPOSITE depends on dataset? If so, the authors need to justify the generalizability and robustness in applying it to other datasets from different tissues.
- 4) In the section of “Stable feature selection”, the authors chose 300 stable features for RNA and ATAC modalities, while top 10% for ADT features. They claimed that the COMPOSITE model is robust to variations in the number of stable features, but no supporting evidence is shown.
- 5) The section on 'Multiplet detection using single-omics data' requires improvement to help readers to understand the method. While the authors adequately explained the joint likelihood and the estimation of the number of cells in each droplet when q is RNA or ATAC, these details are omitted when q is ADT. I recommend the authors to provide these details in an organized manner

to facilitate reader's understanding.

6) Would like to know the computing cost and running time of COMPOSITE, which is important for users who want to use this tool.

Reviewer #2 (Remarks on code availability):

The codes include a detailed README, which is helpful for users. More importantly, I appreciate the authors for providing instructions on linking the output of COMPOSITE with the popular Seurat tool.

Reviewer #3 (Remarks to the Author):

In this manuscript, the authors proposed a novel computational method to find doublets/triplets from single-cell omics data. The method combines multiomics data and several statistical models to infer multiplets. The authors conducted experiments to generate single-cell data with ground-truth multiplets labels and used this data to benchmark proposed methods and existing leading methods. Overall, this is a comprehensive and innovative work. The proposed method is largely different from the existing method and shows its strongness over existing methods. The writing and visualization are high-quality. The software is provided in the open-source platform with clear documentation. I have the following suggestions for the author to improve the manuscript.

1. I am interested in seeing the authors separate the identification of doublets and triplets. Can the authors provide further discussion and analysis to show the difference between doublets and triplets in terms of their detection difficulty and impact on downstream analysis? Why do the authors separate the detection of these two? This perspective is rarely discussed in current literature. In addition, the discussion can be further generalized to quartets, quintets... Is this meaningful in practice? Please give related discussion.

2. The benchmark part could be more comprehensive. First, what is the method's improvement on downstream analysis, such as clustering, DE, trajectory inference, and others? Due to the complexity of those methods, only detection accuracy may not fully reflect the method's real-world impact. Second, the authors can consider using datasets with true labels generated by other techniques. Each technique has its own limitations and false labels. The combination of multiple methods can largely reduce those issues. Third, the authors could consider using simulation to generate multiplets data as supplemental benchmark resources.

3. In Figure 2E, it is better to add a percentage to each circle to show the correct and error rates.

Reviewer #3 (Remarks on code availability):

The authors provided two software implementations on open-source platforms. The software comes with detailed instructions for execution.

REVIEWER COMMENTS

Reviewer #1 (Remarks to the Author):

Summary:

This manuscript presents a new method for single-cell multiplet detection specifically designed for multiomic data, which is an untapped area in the field of bioinformatics. COMPOSITE is able to make more accurate predictions of multiplet status for each droplet within a multimodal dataset compared to existing methods by using 1) stable (i.e., housekeeping) gene expression as features of importance, which they demonstrate as having more distinct distributions between singlets and multiplets when compared to highly variable genes, and 2) weighing the contribution of multiplet calls from each modality on a droplet-by-droplet basis based on consistency, which reduces the effect of noise on multiplet detection. The authors demonstrate the higher performance of their tool compared to existing methods on 17 immune datasets. Overall, I think this method is innovative, statistically-backed, and its good performance is well evidenced on the datasets presented in the manuscript. However, additional consideration should be given to the breadth of existing tools compared against, the use of ATAC-seq-based gene expression inference, and some claims of performance (homotypic doublets and insensitivity to the number of stable features) that are not supported in the data presented.

Major Concerns:

Performance in other tissues besides PBMC / Immune cells

• The authors state that:

“The vast majority of the sorted cells were immune cells, as most non-immune cells were in the excluded dead or apoptotic cell populations.” (lines 656-658)

This means that, in combination with the peripheral blood samples, that effectively all the datasets tested were in immune cells. However, we desperately need methods that can work on more tissues with cell types that may be less defined than those in blood. Perhaps using a cryopreservation technique or sample preparation method would allow for the retention of non-immune cells and would give users confidence that this method would work in their tissues. It is not uncommon for bioinformatics methods to work well in PBMCs and fail on other tissue types, so I think this should be addressed.

R: We appreciate the reviewer highlighting the crucial fact that the performance of bioinformatic tools can vary based on tissue types. To address this concern, we have acquired two additional datasets with experimental truth, featuring a wider range of cell types compared to the ones initially tested in our draft.

The first of these is a colon biopsy sample, specifically enriched for non-immune cells (CD45-) through flow cytometry. It is important to note that protocols for single-cell multiomics experiments on solid tissues are not as developed as those for immune cells, especially in the context of the surface protein modality, which is primarily focused on immune cell marker analysis. Despite this, scRNA-seq has been widely adopted for solid

tissue samples and generally yields data with acceptable qualities. Therefore, we derived scRNA-seq data from this sample, along with ground truth multiplet labels through cell hashing. This enables us to benchmark our method against other popular bioinformatic tools for multiplet detection on a non-immune cell sample on the RNA modality. The single modality version of COMPOSITE outperformed all existing methods, as demonstrated in Figure 1.

Figure 1: Benchmarking of multiplet detection performance on the non-immune cell-enriched ileum scRNA-seq dataset

The second sample is a PBMC DOGMA-seq dataset independently generated by another laboratory at the University of Pittsburgh Medical Center, distinct from our own. In contrast to the T cell-enriched peripheral blood samples detailed in our initial draft, this dataset encompasses a more balanced variety of cell types from blood. Including this sample in our benchmarking process, we aim to showcase the broad applicability of our method. As illustrated in Figure 2, COMPOSITE outperformed existing methods, particularly when supplemented with information from the ADT modality.

Figure 2: Benchmarking of multiplet detection performance on the PBMC DOGMA-seq dataset

The reliability of Signac

• Do you think that the lower goodness of fit and F1 scores for ATAC-seq data may be due to the used of inferred gene expression as the model input instead of some way that would allow you to use the raw data itself? The authors describe these data:

“with scATAC-seq data represented as gene activity inferred by Signac”. (line 134) The authors of Signac acknowledge the limitations of their website in the section describing how to perform this inference: “Note that the activities will be much noisier than scRNA-seq measurements. This is because they represent measurements from sparse chromatin data, and because they assume a general correspondence between gene body/promoter accessibility and gene expression which may not always be the case.” In the vignette quoted above (https://stuartlab.org/signac/articles/pbmc_vignette), the authors of Signac are using the gene expression inference as a method for calling cell type, not to use it as bonafide expression data.

R: We thank the reviewer for the insightful comment regarding the use of inferred gene expression from scATAC-seq data as model input and its potential impact on the performance of our method. We agree that scATAC-seq data, being inherently sparser and noisier compared to scRNA-seq data, pose unique challenges when used to infer gene activity, as noted in the Signac documentation. This inherent noise level is a key factor contributing to the lower performance metrics for scATAC-seq data observed in our study.

We chose to use Signac for gene activity inference because, despite its limitations, it provides a systematic approach to reduce the sparsity of scATAC-seq data, making the data less noisy and more comparable to gene expression data. This preprocessing step is crucial for our modeling framework, as it allows us to integrate scATAC-seq data into

our model more effectively. The authors of Signac caution users about the potential noise in inferred gene activity; however, they also present it as a viable approach for gaining insights from scATAC-seq data that would be challenging to analyze in its raw form. In addition, our methods do not need users to share raw data, especially for scATAC-seq, which contains sensitive DNA information.

To make our model more robust against the noise in the inferred gene activity data from Signac, we have implemented cell-specific modality weights in our model. This approach allows us to dynamically adjust the influence of each data modality for each cell based on the quality of the data. For cells where the inferred gene activity data is relatively less noisy, the modality weights can upweight the contribution of the inferred gene activity data, and conversely, downweight it in cases where the scATAC-seq data is noisier. This nuanced approach enables us to improve the overall robustness of our model.

Lack of demonstration in removing homotypic doublets

• **The introduction and discussion both mention that this method is sensitive to homotypic doublets while the existing methods are not, but I don't see where this has been demonstrated in the manuscript. Specifically in the discussion the authors state:**

“Additionally, the use of stable features endows COMPOSITE with robustness against both heterotypic and homotypic multiplets, whereas many existing methods only demonstrate sensitivity to heterotypic multiplets due to their reliance on highly variable features.” (lines 374-376).

As far as I can tell, the authors do not demonstrate that their tool works to remove homotypic doublets, only stating that it theoretically should based on how the method works. Additionally, they state that when using COMPOSITE, multiplets:

“did not concentrate in any specific clusters, indicating that the unremoved multiplets induced little bias to analytical outcomes.” (lines 332-333)

If these remaining multiplets are homotypic, as they should be given their location in the clusters, then it undermines the claims in the discussion.

R: We sincerely thank the reviewer for this insightful comment regarding COMPOSITE's ability to detect homotypic multiplets. We also thank the reviewer for pointing out that there are still multiplet unremoved within clusters (lines 332-333), which are likely to be homotypic multiplets in theory. We acknowledge that it's hard to demonstrate the removal of homotypic multiplets using real data, since the existing experimental approaches are unable to generate ground truth labels for distinguishing homotypic/heterotypic multiplets.

Nevertheless, we were able to demonstrate COMPOSITE's enhanced ability to remove homotypic multiplets through simulation. For simulation, we utilized datasets from the peripheral blood samples. We simulated twenty general DOGMA-seq datasets with artificial doublets generated by aggregating expression profiles from ground truth singlets. To generate datasets with homotypic doublets, we focused on CD4+ T cells and simulated twenty datasets that only contain artificial homotypic doublets simulated by combining expression profiles of two CD4+ T cells. Specifically, for each of the two settings, we generated two simulated datasets from each of the ten peripheral blood

sample datasets. We then compared COMPOSITE (DOGMA) to scDbFinder (RNA) using the datasets simulated under these two settings. Compared to the general setting (Figure 3A), COMPOSITE (DOGMA) displayed greater superiority over scDbFinder (RNA) in the homotypic setting (Figure 3B). Therefore, the simulation results support COMPOSITE's effectiveness in detecting homotypic multiplets, surpassing existing methods that largely rely on highly variable genes.

In our revised manuscript, we have incorporated these simulation results and moderated our claims regarding COMPOSITE's homotypic multiplet detection capability, due to the absence of experimental ground truth validation.

Figure 3: Simulation results comparing COMPOSITE (DOGMA) and scDbFinder (RNA) in terms of their abilities to detect general multiplets (A) and homotypic multiplets (B).

The benchmarking metrics are F1 scores. In the boxplots, the box spans from the first to third quartile, with the median depicted as a line in the middle. The whiskers extend to 1.5 times the interquartile range (IQR).

Minor Concerns:

More Benchmarking

- I would have liked to see a wider range of other multiplet detection tools tested in addition to the three described in the manuscript, even if it was just a base level comparison of the performance on RNA-seq data alone to get a comparable F1 score.

R: We sincerely appreciate the reviewer's suggestion to broaden the scope of our benchmarking by including a more extensive array of multiplet detection tools. In response, we have expanded our benchmarking to encompass five additional well-regarded methods: DoubletDetection¹, Scrublet², bcdds³, csds³, and hybrid³. These methods were carefully selected based on their relevance and popularity in the field.

Specifically, we performed benchmarking on both the peripheral blood and ileum sample datasets. The results of these analyses, including a side-by-side comparison of F1 scores for each method, are presented in Figure 4 (peripheral blood samples) and Figure 5 (ileum samples).

Figure 4: Boxplots showing the performances (in terms of F1 score) of each multiplet detection method on the in-house peripheral blood samples

In the boxplots, the box spans from the first to third quartile, with the median depicted as a line in the middle. The whiskers extend to 1.5 times the interquartile range (IQR). In the labels of the x-axis, the texts within the parenthesis indicate the modalities of data that were used as input into the corresponding method.

Figure 5: Boxplots showing the performances (in terms of F1 score) of each multiplet detection method on the in-house ileum samples

In the boxplots, the box spans from the first to third quartile, with the median depicted as a line in the middle. The whiskers extend to 1.5 times the interquartile range (IQR). In the labels of the x-axis, the texts within the parenthesis indicate the modalities of data that were used as input into the corresponding method.

Scanpy compatibility

- **I would like to see the tutorial include the preprocessing steps for scanpy (Python) users, not just Seurat (R) users.**

R: We are grateful for the reviewer's constructive suggestion, emphasizing the importance of inclusivity for users with different preferences in data analysis platforms. In response, we have expanded our tutorial to cater not only to Seurat (R) users but also to those who prefer Python for their data analysis needs.

Accordingly, we have included a new section on our GitHub repository dedicated to preprocessing steps using Scanpy (<https://github.com/CHPGenetics/COMPOSITE>). This addition aims to provide clear, step-by-step guidance for Python users, ensuring that the tutorial is comprehensive and accessible to a broader audience. We believe this enhancement will make our method more user-friendly for the community.

• **The authors claim:**

“the COMPOSITE model is robust to variations in the number of stable features” (lines 459-460).

It would be good to demonstrate this in the manuscript.

R: We apologize for lack of clarity in the original manuscript and thank the referee for providing us an opportunity to make it clear. To substantiate our claim regarding COMPOSITE's robustness to the number of stable features selected, we conducted comprehensive evaluations across all 17 datasets, varying the quantity of stable features used. Our results demonstrate a consistent pattern: the performance of the COMPOSITE model initially improves as more stable features are incorporated, eventually stabilizing at a plateau (Figure 6). This pattern indicates that while the inclusion of additional stable features can enhance model performance up to a certain point, beyond this threshold, the marginal gains diminish, underscoring the model's resilience to variations in the number of features selected. We have updated the Figure 6 below as Supplementary Figure 36 in our manuscript.

Figure 6: The relationships between the number of stable features and the single modality COMPOSITE performance

The figure illustrates the relationships between the single modality prediction performance and the number of stable RNA features (A), stable ADT features (B), and stable ATAC features (C). Each boxplot represents the distribution of F1 scores achieved by the single modality prediction using the corresponding number of stable features on all 10 in-house peripheral blood datasets and 7 in-house ileum sample datasets. In the boxplots, the box spans from the first to third quartile, with the median depicted as a line and the mean depicted as a triangle in the middle. The whiskers extend to 1.5 times the interquartile range (IQR).

- The description of the method itself is excellent and very detailed. It might be helpful to define lowercase “x” in lines 479 and 480 for those who are less familiar with the random variable vs. specific value notation.

R: We thank the reviewer for this kindly suggestion aimed at clarifying the notations used in our manuscript. In the updated version, we have explicitly defined lowercase ‘x’, differentiating it from its use as a random variable to ensure that its meaning is clear.

- The authors state that:

“For the PB-1 dataset, all benchmarked methods were successful in eliminating a significant proportion of ground truth multiplets and displayed their importance in practice (Figures 4A-4D; Supplementary Figures 4B-4C)” (lines 297-299)

On the PB-1 dataset, we see UMAP visualizations for the following:

- o COMPOSITE – RNA (Fig 4)**
- o scDbIFinder – RNA (Fig 4)**
- o DoubletFinder – RNA (Fig 4)**
- o COMPOSITE – ADT (Sup. Fig 4)**
- o COMPOSITE – DOGMA (Fig 4)**
- o scDbIFinder – ATAC (Sup. Fig 4)**
- o AMULET – ATAC (Sup. Fig 4)**

Given this thoroughness for all methods and combinations of data, I would like to see COMPOSITE performance on the ATAC seq data alone on a UMAP in Supplementary Figure 4.

• I also want to note that Supplemental Figure 4B is called out before 4A in the text, which was a bit confusing. It might be worth checking the manuscript for other ordering issues such as this for reading clarity.

R: We thank the reviewer for the constructive feedback. We have added a UMAP visualization illustrating the performance of COMPOSITE on the ATAC-seq data alone for the PB-1 dataset. We agree that including this additional data provides a more comprehensive view of COMPOSITE's capabilities across different data modalities and enhances the overall transparency and comparability of our benchmarking efforts.

We appreciate the reviewer's attention to detail and agree that the order in which figures are referenced should facilitate clarity and understanding. We have reviewed and corrected the instance where Supplemental Figure 4B was mentioned before 4A in the text to ensure a logical progression. Additionally, we have carefully checked the entire manuscript for similar ordering issues and made the necessary adjustments to maintain a coherent and logical order throughout. We believe these revisions will improve the overall readability and flow of the manuscript.

Reviewer #2 (Remarks to the Author):

In this work, the authors proposed a statistical model COMPOSITE for multiplet detection in single-cell multiomics data. COMPOSITE exhibited remarkable superiority over existing single-omics multiplet detection methods, based on the 17 trimodal DOGMA-seq experimental datasets. The authors demonstrate that COMPOSITE is an essential tool for integrating cross-modality multiplet signals. Specifically, I have some major comments as below.

1) In the results section, COMPOSITE has been applied to 17 datasets profiled from T-cell-enriched peripheral blood samples and solid tissue samples. However, the benchmarking of multiplet prediction methods is only specifically shown for PB-1

(Figures 2 and 4) and Ileum-1 dataset (Figure 5). What are the benchmarking results on the other datasets?

R: We appreciate the reviewer's attention to the comprehensiveness of our benchmarking analysis across multiple datasets. In the main body of our results section, we primarily showcased the application of COMPOSITE on two specific datasets, PB-1 and Ileum-1, due to space constraints and the desire to maintain clarity and focus on our presentation.

We have generated comprehensive benchmarking analyses on the other 15 datasets. We have included these additional benchmarking results as Supplementary Figures 12-20 and Supplementary Figures 24-29 of our revised manuscript. Specifically, we have generated corresponding UMAP figures that detail the performance of COMPOSITE and other multiplet prediction methods on these datasets.

We believe that the inclusion of these results strengthens our manuscript by demonstrating the robustness and versatility of COMPOSITE. We are grateful for the opportunity to enhance our manuscript with these additional data and thank the reviewer for prompting this comprehensive presentation.

2) In Figure 2E, the contingency table shows the ground truth vs. predicted multiplet status from COMPOSITE. From this table, COMPOSITE tends to detect more doublets and triplets than ground truth. And Figure 2F shows the F1 scores on most datasets are below 0.7. I am quite concerned about the performance of COMPOSITE in detecting false positives of multiplets.

R: We appreciate the reviewer's concern regarding the false positives in multiplet detection by COMPOSITE, as observed in Figure 2E and the F1 scores depicted in Figure 2F. We acknowledge that COMPOSITE appears to identify a higher number of droplets as multiplets compared to the experimental ground truth. While some of these are indeed false positives, it's important to note that some of the seemingly false positive classifications are, upon closer examination, accurate (Figures 4F-4H in the manuscript). This discrepancy may arise from limitations in the experimental ground truth rather than inaccuracies in COMPOSITE's detection capabilities.

The ground truth, derived from cell-hashing, is widely respected as the gold standard in multiplet detection (Figure 1B in the manuscript). Yet, the ground truth is still not perfect, notably its inability to detect multiplets that contain cells with identical hashtag oligos (HTOs). Therefore, the experimental ground truth may lack sensitivity and inadvertently contain false negatives—true multiplets that go unrecognized by the experimental approach.

Consequently, some droplets identified as multiplets by COMPOSITE, contrary to the ground truth, could in fact represent true multiplets overlooked by cell-hashing techniques. This discrepancy highlights COMPOSITE's ability to complement the experimental approaches by identifying multiplets missed in experimental detection processes.

The observed discrepancy contributes to the F1 scores being not optimal. If the ground truth labels were flawless, COMPOSITE's F1 scores would likely improve.

3) As shown in Figure 3C-D, in the peripheral blood samples, COMPOSITE shows quite better performance than existing methods. In contrast, the benchmarking performance on those methods are tend to be comparable on the ileum samples. I am questioning whether the performance of COMPOSITE depends on dataset? If so, the authors need to justify the generalizability and robustness in applying it to other datasets from different tissues.

R: We thank the reviewer for the insightful observations regarding the performance of COMPOSITE when applied to different tissue types. We appreciate the opportunity to further discuss the potential variability in COMPOSITE's efficacy across different datasets.

We acknowledge that, as with many analytical methods, the performance of COMPOSITE may exhibit variation when applied to different sample types. One of the important reasons why the superiority of COMPOSITE is more remarkable in peripheral blood samples compared to the ileum samples is the relative homogeneity of cell types within the blood samples. As we discussed in the manuscript, in theory, COMPOSITE should have better ability on detecting homotypic multiplets (i.e., multiplets formed by the same type of cells) compared to the existing methods, since COMPOSITE leverage stable features (i.e., features that have similar expression levels across different cell types) while the existing methods mainly rely on highly variable features.

To further address the generalizability issue, we have acquired two additional datasets featuring a wider range of cell types compared to the ones in our first draft.

The first of these is a colon biopsy sample. Unlike the ileum samples detailed in our manuscript, which are enriched with immune cells, this sample is specifically enriched for non-immune cells (CD45-) through flow cytometry. It is important to note that protocols for single-cell multiomics experiments on solid tissues are not as developed as those for immune cells, especially for the ADT modality, which is primarily focused on immune cell marker analysis. Despite this, scRNA-seq has been widely adopted for solid tissue samples and generally yields data with acceptable qualities. Therefore, we obtained scRNA-seq data from this sample, along with ground truth multiplet labels through cell hashing. This enables us to benchmark our method against other popular bioinformatic tools for multiplet detection on a non-immune cell sample on the RNA modality. The single modality version of COMPOSITE outperformed all existing methods, as demonstrated in Figure 7.

Figure 7: Benchmarking of multiplet detection performance on a non-immune cell-enriched ileum scRNA-seq dataset

The second sample is a PBMC DOGMA-seq dataset independently generated by another laboratory at the University of Pittsburgh Medical Center, distinct from our own. In contrast to the T cell-enriched peripheral blood samples detailed in our initial draft, this dataset encompasses a more balanced variety of cell types from blood. Including this sample in our benchmarking process, we aim to showcase the broad applicability of our method. As illustrated in Figure 8, COMPOSITE outperformed existing methods, particularly when supplemented with information from the ADT modality.

Figure 8: Benchmarking of multiplet detection performance on a PBMC DOGMA-seq dataset

We have added the two figures above into our manuscript and expanded the discussion within our manuscript to address the generalizability and robustness of COMPOSITE,

especially in relation to its application across diverse tissue datasets. We have included additional discussions in our manuscript to acknowledge that while COMPOSITE is designed to be as robust and generalizable as possible, its performance, like that of any method, can be influenced by the nature of the dataset it is applied to. We believe that COMPOSITE remains a valuable tool for the community, demonstrating significant improvements in performance compared to the single-omics multiplet detection method across the tested datasets. We are dedicated to the ongoing refinement and enhancement of COMPOSITE, adapting and optimizing as new data emerge.

4) In the section of “Stable feature selection”, the authors chose 300 stable features for RNA and ATAC modalities, while top 10% for ADT features. They claimed that the COMPOSITE model is robust to variations in the number of stable features, but no supporting evidence is shown.

R: We apologize for lack of clarity in the original manuscript and thank the referee for providing us an opportunity to make it clear. To substantiate our claim regarding COMPOSITE's robustness to the number of stable features selected, we conducted comprehensive evaluations across all 17 datasets, varying the quantity of stable features used. Our results demonstrate a consistent pattern: the performance of the COMPOSITE model initially improves as more stable features are incorporated, eventually stabilizing at a plateau (Figure 9). This pattern indicates that while the inclusion of additional stable features can enhance model performance up to a certain point, beyond this threshold, the marginal gains diminish, underscoring the model's resilience to variations in the number of features selected. We have updated the Figure 9 below as Supplementary Figure 36 in our manuscript.

Figure 9: The relationships between the number of stable features and the single modality COMPOSITE performance

The figure illustrates the relationships between the single modality prediction performance and the number of stable RNA features (**A**), stable ADT features (**B**), and stable ATAC features (**C**). Each boxplot represents the distribution of F1 scores achieved by the single modality prediction using the corresponding number of stable features on all 17 in-house datasets. In the boxplots, the box spans from the first to third quartile, with the median depicted as a line and the mean depicted as a triangle in the middle. The whiskers extend to 1.5 times the interquartile range (IQR).

5) The section on 'Multiplet detection using single-omics data' requires improvement to help readers to understand the method. While the authors adequately explained the joint likelihood and the estimation of the number of cells in each droplet when q is RNA or ATAC, these details are omitted when q is ADT. I recommend the authors to provide these details in an organized manner to facilitate reader's understanding.

R: We thank the reviewer for the thorough review and the constructive feedback regarding the readability of the methods section. We have revised this section to include a more detailed explanation of the approach we employed for multiplet detection on ADT data modality. Like the descriptions provided for RNA and ATAC data, we now

elucidate the joint likelihood estimation process and how we estimate the number of cells in each droplet based on ADT data.

6) Would like to know the computing cost and running time of COMPOSITE, which is important for users who want to use this tool.

R: We are grateful to the reviewer for emphasizing the significance of showing the computing cost and running time associated with the COMPOSITE method. By leveraging GPU acceleration, the COMPOSITE method can efficiently perform multiplet detection for a single 10X Chromium well within minutes, as demonstrated in Figure 10 below. We have updated the Figure 10 as Supplementary Figure 34 in our manuscript. Multiple-well data will be processed in parallel, so our streamlined approach should be applied to any droplet-based experiments without concern of computational burden.

Figure 10: The relationships between the number of droplets in the dataset and the computational time of COMPOSITE

The figure illustrates the relationships between the single modality computational time of COMPOSITE and the number of droplets in the dataset respectively for the RNA modality (A), the ADT modality (B), and the ATAC modality (C). Each boxplot represents the distribution of computational time for COMPOSITE using the corresponding number of randomly selected droplets from the 9 in-house datasets with over 16000 droplets. 300 stable features were used for the RNA and ATAC modalities,

and 16 stable features were used for the ADT modality. In the boxplots, the box spans from the first to third quartile, with the median depicted as a line and the mean depicted as a triangle in the middle. The whiskers extend to 1.5 times the interquartile range (IQR). The GPU used for computing was an NVIDIA A100 PCIe.

Reviewer #2 (Remarks on code availability):

The codes include a detailed README, which is helpful for users. More importantly, I appreciate the authors for providing instructions on linking the output of COMPOSITE with the popular Seurat tool.

R: We appreciate the comments. We have further polished the GitHub by expanding our tutorial to cater not only to Seurat (R) users but also to those who prefer Python for their data analysis needs.

Reviewer #3 (Remarks to the Author):

In this manuscript, the authors proposed a novel computational method to find doublets/triplets from single-cell omics data. The method combines multiomics data and several statistical models to infer multiplets. The authors conducted experiments to generate single-cell data with ground-truth multiplets labels and used this data to benchmark proposed methods and existing leading methods. Overall, this is a comprehensive and innovative work. The proposed method is largely different from the existing method and shows its strongness over existing methods. The writing and visualization are high-quality. The software is provided in the open-source platform with clear documentation. I have the following suggestions for the author to improve the manuscript.

1. I am interested in seeing the authors separate the identification of doublets and triplets. Can the authors provide further discussion and analysis to show the difference between doublets and triplets in terms of their detection difficulty and impact on downstream analysis? Why do the authors separate the detection of these two? This perspective is rarely discussed in current literature. In addition, the discussion can be further generalized to quartets, quintets... Is this meaningful in practice? Please give related discussion.

R: We appreciate the reviewer's interest in our approach to differentiating between the detection of doublets and triplets, and the request for a deeper discussion on their distinctions. This differentiation is indeed a nuanced aspect of our work that merits further discussion, given its potential implications for the accuracy of multiplet detection and the integrity of transcriptomic profiles.

Doublets and triplets, while both contributing to artificial transcriptomic signatures, exhibit distinct gene expression distributions. Recognizing these differences allows our model to more accurately mirror biological realities, thereby enhancing its predictive performance. Such differentiation is rare in current literature, largely due to the modeling challenges

and diminishing frequency of larger multiplets. However, larger multiplets are more common in high-throughput single-cell multiomics experiments with higher cell loading densities, making their detection vital for ensuring the robustness of multiplet identification methods. In our modeling process, distinguishing between doublets and triplets—and, by extension, quartets, quintets, and beyond—serves to refine the model fitting to the multiplet landscape.

In collective response to this comment and your second comment, we have conducted relevant simulation to demonstrate that separately modeling doublets and triplets is especially helpful for enhancing the model performance in the settings where the proportion of triplets is large. The details are in our response to your second comment. The practical relevance of distinguishing beyond triplets, such as quartets and quintets, depends on the specifics of the experimental protocols, like cell loading density, which affect the likelihood of forming larger multiplets.

To this end, we have updated our manuscript with a more detailed discussion on the distinctions between doublets, triplets, and larger multiplets.

2. The benchmark part could be more comprehensive. First, what is the method's improvement on downstream analysis, such as clustering, DE, trajectory inference, and others? Due to the complexity of those methods, only detection accuracy may not fully reflect the method's real-world impact. Second, the authors can consider using datasets with true labels generated by other techniques. Each technique has its own limitations and false labels. The combination of multiple methods can largely reduce those issues. Third, the authors could consider using simulation to generate multiplets data as supplemental benchmark resources.

R: We sincerely thank the reviewer for providing these insightful and very helpful suggestions. Since most of the existing methods are effective methods and can remove a significant amount of multiplets, the differences between their impact on downstream analysis are more nuanced and not as easy to illustrate as the detection accuracy. Nevertheless, we recognize the importance of evaluating the real-world impact of our method beyond mere detection accuracy. To this end, we have expanded our analysis to include the impacts of COMPOSITE on downstream analysis, particularly focusing on clustering and trajectory inference, which serve as foundational steps for further analyses, such as differential expression (DE) analysis.

We use the PB-1 dataset as an example for illustration. In the clustering analysis, before multiplet removal, clusters 3, 4, 5, and 6 all have significant proportions of multiplets (Figures 11A, 11B), potentially biasing the downstream analysis. After multiplet removal with COMPOSITE (DOGMA), none of the identified clusters contain significant amount of multiplets, thus reducing potential biases in subsequent analyses (Figures 11C, 11D).

Figure 11: The impact of COMPOSITE (DOGMA) multipler removal on the PB-1 dataset clustering results

A: UMAP plot showing the clustering results before multipler removal

B: UMAP plot showing the ground truth multipler status of each droplet before multipler removal

C: UMAP plot showing the clustering results after multipler removal with COMPOSITE (DOGMA)

D: UMAP plot showing the ground truth multipler status of each droplet after multipler removal with COMPOSITE (DOGMA)

For trajectory inference, we extracted CD4+ T cells based on ADT expressions in scenarios both before and after multipler removal with COMPOSITE (DOGMA) (Figures 12A, 12B). Subsequently, we performed trajectory inference independently for each scenario using Monocle 3 (Figures 12C, 12D). Before the removal of multiplers, trajectory analysis revealed two branches, marked by red circles in Figure 12C, extending into clusters identified as multiplers according to experimental ground truth (Figures 12C, 12E). In contrast, after applying COMPOSITE (DOGMA) for multipler removal, these aberrant branches were no longer present, indicating a cleaner trajectory inference free from the influence of multiplers (Figures 12D, 12F).

Figure 12: The Impact of Multiplet Removal by COMPOSITE (DOGMA) on Trajectory Inference for CD4+ T Cells in the PB-1 Dataset

A: UMAP plot showing the CD4 surface protein marker expression of each droplet before multiplet removal.

B: UMAP plot showing the CD4 surface protein marker expression of each droplet after multiplet removal with COMPOSITE (DOGMA).

C: UMAP plot showing the trajectory inference on annotated CD4+ T cells before multiplet removal. The red circles mark the branches of the trajectory that extend into the multiplet clusters.

D: UMAP plot showing the trajectory inference on annotated CD4+ T cells after multiplet removal with COMPOSITE (DOGMA).

E: UMAP plot showing the ground truth multiplet status of the droplets that are annotated to be CD4+ T cells before multiplet removal.

F: UMAP plot showing the ground truth multiplet status of the droplets that are annotated to be CD4+ T cells after multiplet removal with COMPOSITE (DOGMA).

We sincerely appreciate the reviewer's suggestion to enhance our validation process by generating ground truth labels using various techniques. Currently, our dataset primarily relies on cell-hashing for ground truth labels, which we believe provides the most reliable labels currently accessible to our team. Unfortunately, due to logistical constraints, we are unable to implement additional experimental techniques to generate alternative ground truth labels at this time. To our best knowledge, there is no publicly available single-cell multiomics dataset with ground truth multiplet labels. We are dedicated to the ongoing refinement and enhancement of COMPOSITE, adapting and optimizing as new data emerge. However, recognizing the importance of comprehensive benchmarking, we concur with the reviewer that supplementing our analysis with simulation results can offer a more robust evaluation, particularly in offering the flexibility to create datasets with varying doublet and triplet proportions. For simulation, we first removed all ground truth multiplets from the peripheral blood datasets, and then simulated artificial multiplets by combining expression profiles from ground truth singlets within the datasets. We conducted simulations across scenarios with artificial doublet rates from 5% to 30% and triplet rates from 2% to 8%, generating 20 simulated datasets for each scenario. Specifically, under each scenario, two simulated datasets were generated from each of the ten peripheral blood sample datasets. Our results, depicted in Figures 13A-13D, consistently show superior performance of COMPOSITE (DOGMA) over scDbfFinder (RNA), particularly as the rate of triplets increases. These results suggest that by distinguishing doublets and triplets in the modeling process, COMPOSITE can effectively handle the datasets with significant proportions of larger multiplets, such as triplets.

Figure 13: Benchmarking COMPOSITE (DOGMA) with scDbFinder (RNA) on simulated datasets with various doublet and triplet rates

A: Doublet rates ranging from 10% to 30% and triplet rates fixed to 2%

B: Doublet rates ranging from 10% to 30% and triplet rates fixed to 4%

C: Doublet rates ranging from 10% to 30% and triplet rates fixed to 6%

D: Doublet rates ranging from 10% to 30% and triplet rates fixed to 8%

The benchmarking metrics are F1 scores. In the boxplots, the box spans from the first to third quartile, with the median depicted as a line in the middle. The whiskers extend to 1.5 times the interquartile range (IQR).

3. In Figure 2E, it is better to add a percentage to each circle to show the correct and error rates.

R: We are thankful for the reviewer's insightful recommendation to augment the clarity of Figure 2E. In accordance with this guidance, we have updated the figure to include precise percentages for each prediction category (Figure 14). We wish to clarify that the classification of true doublets as triplets, as well as the classification of true triplets as doublets, does not adversely affect downstream analysis since both categories are intended to be excluded. Therefore, such classifications are not considered as errors. Consequently, we have avoided from labeling the categories in the figure as either correct or erroneous to prevent potential misinterpretation.

Figure 14: Contingency table comparing ground truth vs. predicted multiplet status from COMPOSITE.

The numbers on each intersection point of the grids represents the number and proportion of droplets that belong to the corresponding category, and the sizes of the dots on the grid intersections represent the magnitudes of the corresponding numbers.

Reviewer #3 (Remarks on code availability):

The authors provided two software implementations on open-source platforms. The software comes with detailed instructions for execution.

R: We appreciate the comments. We have further polished GitHub by expanding our tutorial to cater not only to Seurat (R) users but also to those who prefer Python for their data analysis needs.

Reference

1. Gayoso, A.S., Jonathan DoubletDetection. *Zenodo* (2018).
2. Wolock, S.L., Lopez, R. & Klein, A.M. Scrublet: Computational Identification of Cell Doublets in Single-Cell Transcriptomic Data. *Cell Syst* **8**, 281-291 e289 (2019).
3. Bais, A.S.K., Dennis scds: computational annotation of doublets in single-cell RNA sequencing data. *Bioinformatics* **36**, 8 (2019).

Reviewer #1 (Remarks to the Author):

The authors adequately addressed all my concerns and I have no further concerns.

Reviewer #2 (Remarks to the Author):

The authors have done sufficient experiments to address my previous comments. Now I have no more concerns about this manuscript.

Reviewer #3 (Remarks to the Author):

The authors have addressed my concerns.

Reviewer #3 (Remarks on code availability):

The authors provide working code